# Potential of Human Hemoglobin as a Source of Bioactive Peptides: Comparative Study of Enzymatic Hydrolysis with Bovine Hemoglobin and the Production of Active Peptide α137–141

**DOI:** 10.3390/ijms241511921

**Published:** 2023-07-25

**Authors:** Ahlam Outman, Barbara Deracinois, Christophe Flahaut, Mira Abou Diab, Bernard Gressier, Bruno Eto, Naïma Nedjar

**Affiliations:** 1UMR Transfrontalière BioEcoAgro N°1158, Institut Charles Viollette, National Research Institute for Agriculture, Food and the Environment-Université Liège, UPJV, YNCREA, Université Artois, Université Littoral Côte d’Opale, Université Lille, F-59000 Lille, France; ahlam.outman.etu@univ-lille.fr (A.O.); barbara.deracinois@univ-lille.fr (B.D.); christophe.flahaut@univ-artois.fr (C.F.); mira.abou-diab.1@ulaval.ca (M.A.D.); naima.nedjar@univ-lille.fr (N.N.); 2Laboratoires TBC, Laboratory of Pharmacology, Pharmacokinetics and Clinical Pharmacy, Faculty of Pharmaceutical and Biological Sciences, University of Lille, 3, rue du Professeur Laguesse, F-59000 Lille, France; 3Laboratory of Pharmacology, Pharmacokinetics and Clinical Pharmacy, Faculty of Pharmaceutical and Biological Sciences, University of Lille, 3, rue du Professeur Laguesse, F-59000 Lille, France; gressier.bernard@univ-lille.fr

**Keywords:** bovine hemoglobin, human hemoglobin, bioactive peptides, neokyotorphin

## Abstract

Cruor, the main component responsible for the red color of mammalian blood, contains 90% haemoglobin, a protein considered to be a rich source of bioactive peptides. The aim of the present study is to assess the potential of human hemoglobin as a source of bioactive peptides, compared with bovine hemoglobin, which has been extensively studied in recent years. More specifically, the study focused on the α137–141 fragment of bovine haemoglobin (TSKYR), a small (653 Da) hydrophilic antimicrobial peptide. In this work, the potential of human hemoglobin to contain bioactive peptides was first investigated in silico in comparison with bovine hemoglobin-derived peptides using bioinformatics tools. The blast results showed a high identity, 88% and 85% respectively, indicating a high similarity between the α and β chains. Peptide Cutter software was used to predict cleavage sites during peptide hydrolysis, revealing major conservation in the number and location of cleavage sites between the two species, while highlighting some differences. Some peptides were conserved, notably our target peptide (TSKYR), while others were specific to each species. Secondly, the two types of hemoglobin were subjected to similar enzymatic hydrolysis conditions (23 °C, pH 3.5), which showed that the hydrolysis of human hemoglobin followed the same reaction mechanism as the hydrolysis of bovine hemoglobin, the ‘zipper’ mechanism. Concerning the peptide of interest, α137–141, the RP-UPLC analyses showed that its identification was not affected by the increase in the initial substrate concentration. Its production was rapid, with more than 60% of the total α137–141 peptide production achieved in just 30 min of hydrolysis, reaching peak production at 3 h. Furthermore, increasing the substrate concentration from 1% to 10% (*w*/*v*) resulted in a proportional increase in α137–141 production, with a maximum concentration reaching 687.98 ± 75.77 mg·L^−1^, approximately ten-fold higher than that obtained with a 1% (*w*/*v*) concentration. Finally, the results of the UPLC-MS/MS analysis revealed the identification of 217 unique peptides in bovine hemoglobin hydrolysate and 189 unique peptides in human hemoglobin hydrolysate. Of these, 57 peptides were strictly common to both species. This revealed the presence of several bioactive peptides in both cattle and humans. Although some had been known previously, new bioactive peptides were discovered in human hemoglobin, such as four antibacterial peptides (α37–46 PTTKTYFPHF, α36–45 FPTTKTYFPH, α137–141 TSKYR, and α133–141 STVLTSKYR), three opioid peptides (α137–141 TSKYR,β31–40 LVVYPWTQRF,β32–40, VVYPWTQRF), an ACE inhibitor (β129–135 KVVAGVA), an anticancer agent (β33–39 VVYPWTQ), and an antioxidant (α137–141 TSKYR). To the best of our knowledge, these peptides have never been found in human hemoglobin before.

## 1. Introduction

Blood, whether it comes from animal slaughterhouses or human blood donations, is a coproduct very rich in proteins. Indeed, the food and/or pharmaceutical industries are now turning to the valorization of these raw materials and seeking to develop products with high added value. Only 30% of blood is valorized [1] in slaughterhouses, while the rest is considered waste, although it is rich in proteins and constitutes an ideal substrate for proteolysis. It contains about 17.8% in cattle and 20–25% of all proteins encoded by the human genome are known to be secreted there [2]. The potential of these many co-products remains underutilized. These raw materials currently constitute new resources with a high potential for valorization.

Bovine blood has been very successful commercially both in relation to its collection and processing, unlike most animal co-products [3]. Its wide availability and relative homogeneity make it a promising case study for the production and valorization of bioactive compounds [4,5]. Moreover, in humans, this body fluid is also available in blood transfusion centers or collected through blood donation, unlike saliva or tears.

Blood is essentially subdivided into two parts after centrifugation in vertebrates. The colorless part, plasma, represents about 60% of the volume of blood and can be used in pharmacy because of its richness in thrombins, fibrinogen, or serum albumin [4,6], The part that provides the red color of blood and represents up to 40% of the blood volume is called “cruor” and it contains mainly hemoglobin (90%).

In recent studies, the role of these dietary proteins as a source of biologically active peptides is widely described. Hemoglobin represents more than half of the proteins in many databases that reference the various activities on active peptides [5,6].

Hemoglobin (H) in vertebrates is a tetramer of two α-chains and two β-globin chains, each of which contains a prosthetic heme [7].

For example, the hydrolysis of bovine hemoglobin by pepsin is the source of many peptides with several biological activities, such as opioids [8,9], hematopoietic [10], and antihypertensivity [11]. Nevertheless, the most described aspect remains the antimicrobial activity [12,13,14]. According to a study by Przybylski et al. [15]. α137–141 peptide (also known as neokyotorphin, 653 Da, Pi of 10.5) is an antimicrobial peptide that has been identified as a natural preservative for meat during storage or distribution. This peptide has been shown to reduce rancidity time by 60% and inhibit microbial growth after 14 days in the refrigerator. The study found that α137–141 peptide had a similar effectiveness to butylated hydroxytoluene (BHT), which is commonly used as a meat preservative. This discovery has the potential to revolutionize the food industry and provide a natural alternative to synthetic preservatives.

The most commonly used procedure to generate α137–141 peptide from hemoglobin is enzymatic hydrolysis [11,13,16]. Yet, in the same way as bovine hemoglobin, human hemoglobin has a protein richness that is too high. Indeed, as mentioned above, it is an ideal substrate for proteolysis, generating bioactive peptides that could turn out to be an ingenious strategy for the process of discovery and development of natural potential candidates or molecularly targeted drug candidates for the substitution of synthetic compounds [17,18].

The laboratory has developed a patented strategy to obtain active peptides through the hydrolysis of hemoglobin. Indeed, obtaining the latter remains extremely sensitive because it remains an irreversible process generating varied peptide populations according to plural operating parameters (pH, temperature, etc.) which themselves influence the selectivity and specificity of the enzyme for its substrate [19,20].

In this perspective, the current work aims to demonstrate the feasibility of enzymatic hydrolysis of human hemoglobin to produce α137–141. To do so, a comparative study of human hemoglobin versus bovine hemoglobin through a bioinformatics study was first performed. Then, the peptic hydrolysis of human hemoglobin was compared with the peptic hydrolysis of bovine hemoglobin, the laboratory model, in order to define the similarities in the generated peptides and the presence of bioactive peptides.

This study had three specific objectives. The first was to identify the enzymatic mechanism involved in the hydrolysis of human hemoglobin. The second objective was to quantify the production of α137–141. Finally, the third objective was to characterize andidentify the population of peptides and more particularly the active peptides resulting from the hydrolysis process.

## 2. Results

### 2.1. In Silico Comparative Analysis of Bovine (Bos taurus) and Human (Homo sapiens) Hemoglobin

The scientific name for humans is *Homo sapiens*. For the domestic cattle, the scientific name is *Bostaurus*. The two alpha and beta chains of bovine or human hemoglobin are identical. The blast in Figure 1 is thus performed with two programs, (1) blastp from NCBI [21] and (2) Local Similarity Program (SIM) from ExPASy https://web.expasy.org/sim/ [22], in orderto compare the *Homo sapiens* (GI: 4504345) and *Bostaurus* (GI: 116812902) alpha and beta hemoglobin.

The two α-chains of bovine and human hemoglobin each consist of 142 amino acids, so there are no gaps when aligning the sequences. The Score obtained with blastp is 252, and it corresponds to the statistical measure of the validity of the alignment. A high score indicates that the two sequences compared were similar. Below 50, the result is considered unreliable. The expect value (e-value) corresponds to the value at which we consider that the matching is not due to chance. The closer the value is to zero, the better the result. Here, it was 7.10^−93^, so the result is correct.

The alpha chains *Bos taurus* and *Homo sapiens* (Figure 1A, SIM program) have many similarities in their amino acid sequences: 125 of the 142 are strictly aligned; the value of the identity obtained is 88%. Considering the similar amino acids, this positive value increased to 92%. This alignment of hemoglobin alpha chains allows us to observe that the peptides GAEALER (hematopoietic) and TSKYR (opioid, antimicrobial, and antioxidant) are conserved [23,24,25,26]. Contrarily, the peptides LANVST (Analgesic and Bradykinin Potentiator), ANVST (DPP-IV Inhibitor), and KLLSHSL (Antihypertensive) are only present in *Bos taurus* hemoglobin [11,27].

The same study was performed to align the amino acid sequences of the two β chains of hemoglobin *Bos taurus* (GI: 27819608) and *Homo sapiens* (GI: 4504349). The β-chain of *Bos taurus* contains 145 amino acids compared with 147 for the β-chain of *Homo sapiens*. After sequence alignment, 147 amino acids were compared. The score obtained using the blastp (254), expectation (2.10^−93^), and gap values (0%) allowed us to conclude that the alignment result is correct. The identity between these two β-chains is 85% and rises to 91% when similar amino acids are taken into account (Figure 1B, SIM program). The VVYP (lipid lowering) and LVVYPWTQRFF (opioid) peptides are conserved, as shown by this alignment of hemoglobin β-chains [28,29]. In addition, it should be noted that the human hemoglobin peptide VAGVANALAHKYH (coronary constrictor) was modified to the peptide VAGVANALAHRYH in the β-chain of bovine hemoglobin. Furthermore, the STPDA peptide was modified from the STADA (bacterial growth stimulant) peptide of *Bos taurus* hemoglobin in the chain of human hemoglobin [30].

The same study was performed to align the amino acid sequences of the two β chains of hemoglobin *Bos taurus* (GI: 27819608) and *Homo sapiens* (GI: 4504349). The β-chain of *Bos taurus* contains 145 amino acids compared with 147 for the β-chain of *Homo sapiens.* After sequence alignment, 147 amino acids were compared. The score obtained using the blastp (254), expectation (2.10^−93^), and gap values (0%) allowed us to conclude that the alignment result is correct. The identity between these two β-chains is 85% and rises to 91% when similar amino acids are taken into account (Figure 1B, SIM program). The VVYP (lipid lowering) and LVVYPWTQRFF (opioid) peptides are conserved, as shown by this alignment of hemoglobin β-chains [28,29]. In addition, it should be noted that the human hemoglobin peptide VAGVANALAHKYH (coronary constrictor) was modified to the peptide VAGVANALAHRYH in the β-chain of bovine hemoglobin. Furthermore, the STPDA peptide was modified from the STADA (bacterial growth stimulant) peptide of *Bos taurus* hemoglobin in the chain of human hemoglobin [30].

### 2.2. Bioinformatics Approach to Predict Peptides Derived from the Pepsic Hydrolysis of Bovine Hemoglobin Bos taurus and Human Hemoglobin Homo sapiens

The alpha and beta chain sequences of the *Bos taurus* hemoglobin (GI: 116812902; GI: 27819608) and human hemoglobin (GI: 4504345; GI: 4504349) were submitted to the Peptide Cutter software (ExPASy). Figure 2 illustrates the predicted cleavage sites for peptic hydrolysis at pH higher than 2 for the alpha chains of two hemoglobin. They were 43 and 39, respectively. When the comparable amino acids were considered, Blast reported a 92% identity between the alpha chains of *Homo sapiens* and the *Bos taurus*. Peptide Cutter identified four distinct cleavage sites, which are indicated by red arrows. The bottom arrows point to several different cleavage sites. Note that for the alpha chain of human hemoglobin, one of the missing cleavages is the one that obtains the peptide VAAA (DPP-IV inhibitor) [11,27]. Through prediction, this peptide will not be recovered during the hydrolysis of human hemoglobin.

The cleavage sites predicted by Peptide Cutter, for the peptic hydrolysis, at a pH higher than 2, of the beta chains are presented in Figure 2B. Here, 41 and 42 potential cleavages were identified for the beta chains of bovine *Bos taurus* and human *Homo sapiens* hemoglobin, respectively. However, we note that the cleavages are not identical: 10 different cleavages sites were identified by the bottom arrows. Considering the similar amino acids, the identity between these two beta chains *Bos taurus* and *Homo sapiens* given by Blast increased to 89%. For example, for the bovine hemoglobin beta chain, one of the cleavages resulted in a cut in the bacterial growth stimulating peptide STADA [30], which was not the case for the *Homo sapiens* beta chain. Through prediction, this peptide will not be found during the hydrolysis of human hemoglobin. On the other hand, the peptides VVYP (lipid-lowering) and LVVYPWTQRFF (opioid) are conserved during peptic hydrolysis at a pH above 2 according to Peptide Cutter [11,30,31].

### 2.3. Enzymatic Kinetics and Mechanism of Action Involved

#### 2.3.1. Determination of the Degree of Hydrolysis

The primary objective of this study was to draw a comparison between the hydrolysis of bovine hemoglobin and the hydrolysis of human hemoglobin. It would thus be interesting to compare the different degrees of hydrolysis (DH)under the applied hydrolysis conditions (23 °C, pH 3.5) and at a C_BH_; CHH of 1% (*w*/*v*).

The degree of hydrolysis (DH) did not show a linear relationship overtime in either species (Figure 3). As time progressed, the evolution of DH was identical, which is illustrated in Figure 3, as both curves have the same shape. The DH was 2, 3, 4, 5, 6, 8, and 10% when the hydrolysis time was 2.5 min, 5 min, 15 min, 30 min, 1 h, 2 h, and 3h under these conditions. This is indeed an indicator of the progress of the enzymatic reaction.

#### 2.3.2. Identification of α137–141 by Waters Software

The identification of α137–141 peptide was performed with the Empower 3 software (Version 3 Waters). Several parameters were then considered: the retention time of the peptide, its real-time spectrum, and the estimation of the peak purity, all three of which have been described in detail previously (see Section 4.6.2). This method has been shown to be effective at identifying and quantifying α137–141 in bovine cruor, as previously published in the literature [15].

The standard α137–141 had a retention time of approximately 5.3–5.5 min. Next, the peptide from the human and bovine hydrolysate was analyzed by comparing them to the standard spectrum. With the four parameters (MA, MT, PA, and PT) and retention time, the peak at a little over 5min was identified as α137–141. All measurements showed that PA was lower than PT and MA was lower than MT, highlighting good purity of the peak and the similarity of the spectrum to α137–141.

In addition, a comparison was made for the second derivative spectra (Figure 4) between standard α137–141 peptide (a) and the obtained α137–141 peptide from the hydrolysis of bovine hemoglobin (b) and human hemoglobin (c) using RP-UPLC. This analysis allowed for the estimation of the presence or absence of aromatic amino acid in the peak [32]. In this case, the presence of a tyrosine was assessed, which is in agreement with the sequence of the α137–141 peptide [28]. The specific maxima at 288.7 nm indicated the presence of this amino acid. Their excellent similarity thus allowed for the peak at a retention time of 5 min to be identified within the human and bovine hydrolysates as α137–141.

#### 2.3.3. Reaction Mechanism of the Enzymatic Hydrolysis of Human Hemoglobin by Pepsin

To understand the mechanism of enzymatic hydrolysis of human hemoglobin, the RP-UPLC chromatographic profiles obtained as a function of time were analyzed. Figure 5 presents the chromatograms obtained for identical hydrolysis times with bovine hemoglobin and human hemoglobin, both for a C_BH_ and C_HH_ equal to 1% (*w*/*v*) at 23 °C, pH 3.5. At first sight, it can be noticed that for a given DH, the chromatograms were almost the same. This first element ensured that the kinetics of obtaining peptides from bovine hemoglobin human hemoglobin were subject to the same reaction mechanisms.

Before the addition of pepsin, three peaks were found for T_0_: α-chain, β-chain, and heme. The hemoglobin subunits were quickly hydrolyzed once the enzyme was added, resulting in intermediate peptides that were then converted into final peptides, which were generally more hydrophilic and smaller than the intermediate peptides. This enzymatic process, known as the ’zipper’ mechanism, was first described by Linderstrom-Lang in 1953 and has since been widely studied in the scientific literature [15,20,33]. This is explained by the fact that hemoglobin (bovine or human) was initially in its native form, called “globular”. The latter was denatured by lowering the pH to 3.5 and adding 2 M HCl. It is thus found in an acidic environment in its “expanded” form. This favored the attack of the alpha and beta chains by the enzyme. This favored the enzymatic mechanism zipper, to the detriment of the mechanism ‘one-by-one’, observed for a higher pH [15]. At the end of the reaction, a set of peptides in solution were analyzed.

It is important to note that pepsin does not have the same cutting specificity as trypsin. Consequently, the reaction conditions, including pH and temperature, can have a significant impact on the production of peptide fragments. These variations in reaction conditions can lead to differences in the products obtained during enzymatic hydrolysis. It is thus crucial to control these parameters precisely in order to obtain consistent and reproducible results.

Under the hydrolysis conditions used here (23 °C; pH 3.5; C_BH_ and C_HH_ of 1% *w*/*v*), the production of α137–141 was rapid. This peptide appeared early, a mere 2 min after the start of hydrolysis, i.e., DH = 1% and its concentration increased throughout the hydrolysis, so α137–141 accumulated during the process. It was thus a final peptide: it was not hydrolyzed into a smaller peptide, in agreement with Lignot, Froidevaux, Nedjar-Arroume, and Guillochon [8,13], who specified that α137–141 was a final peptide. For the chromatographic profiles (Figure 5), other fractions were observed containing numerous intermediate peptides that were progressively hydrolyzed to final peptides throughout the hydrolysis of hemoglobin.

In conclusion, the hydrolysis of human hemoglobin showed the same enzymatic mechanism as the hydrolysis of bovine hemoglobin. It is indeed a potential source to produce active peptides such as antimicrobial peptides.

### 2.4. Obtaining the Active Peptide α137–141 during the Peptide Hydrolysis of Human and Bovine Hemoglobin

#### 2.4.1. Effect of Increasing the Initial Concentration of Bovine Hemoglobin on the Enzymatic Obtention of Active Peptide α137–141

As the peak of the peptide of interest, α137–141, was pure, the determination of the production of this peptide during hydrolysis could thus be performed by RP-UPLC, while respecting the previously defined parameters (see Section 4). It was necessary to increase C_BH_ and C_HH_ in order to evaluate the production of the peptide on a larger scale and to verify the production of the peptide at sufficiently high initial concentrations. For this purpose, the assay was performed for all C_BH_ and C_HH_ studied from the hemoglobin (1, 2, 8, and 10%, *w*/*v*) and for all DHs (0 to 10%). The chromatographic profiles of bovine and human hydrolysis for all of the tested C_BH_ and C_HH_ are shown, respectively, in Appendix A. It is observed that whatever the concentration of human/bovine hemoglobin initially used, the hydrolysis mechanism, of the zipper type, remained identical. Moreover, the RP-UPLC analyses indicated that the identification of the peptide was not impacted by the increase in C_HB_/C_HH_.

These measurements not only verified the production of α137–141 at a higher hemoglobin concentration, but they also confirmed the ability of pepsin to hydrolyze at high substrate concentrations. However, a concentration higher than 10% (*w*/*v*) could be problematic because, in this case, the initial medium was no longer homogeneous if the solubilization of hemoglobin was performed simply in water, without the addition of salt.

#### 2.4.2. Quantification of the Production of the Active Peptide α137–141

Figure 6 shows the kinetics of α137–141 peptide as a function of DH and the different C_BH_ and C_HH_ tested. Looking at the peptic hydrolysis of human hemoglobin, at a C_HH_ of 1% (*w*/*v*) and a DH of 10%, Cα137–141 was 69.70 ± 6.76 mg L^−1^. Doubling the initial C_HH_ to 2% (*w*/*v*), Cα137–141 was also increased by a factor of 2 to reach a final concentration of 138.73 ± 6.57 mg·L^−1^. With 8% (*w*/*v*) C_HH_, the hydrolysis yielded a final α137–141 concentration equal to 562.03 ± 106.26 mg·L^−1^, i.e., a concentration approximately eight times higher than that obtained at 1% (*w*/*v*). Finally, the hydrolysis with C_HH_ equal to 10% (*w*/*v*) yielded about ten times that of 1% (*w*/*v*), i.e., C_α137–141_ = 687.98 ± 75.77 mg·L^−1^. Exactly the same findings were made in the pepsic hydrolysis of bovine hemoglobin. This is in accordance with what was described by Przybylski, R et al. (2016). These results once again affirm that the enzyme maintains the same hydrolysis mechanism for human hemoglobin and bovine hemoglobin.

In addition, it is noted that the mechanism was rapid because α137–141 appeared very early. Regardless of the C_HH/_C_BH_ tested, the shape of the kinetics remained the same.

The appearance of α137–141 slowed down and the kinetics appeared to reach a plateau. These data prove that α137–141 was a final peptide under these hydrolysis conditions forboth humans and bovines [8,15].

In an effort to complement the previous observations, it was of interest to evaluate the yield of α137–141 production for both species. Figure 7 shows the α137–141 production yields for all of the C_HH_/C_BH_ tested. The yield was calculated using the following formula:$$R\alpha 137-141=\frac{C\alpha 137-141}{C\alpha 137-141\max}*100$$
where Cα137–141 is the α137–141 concentration measured in UPLC for a given DH and C_BH/HH_ and Cα137–141max is the theoretical maximum α137–141 concentration that can be obtained for a given C_BH/HH_.

At a final DH of 10%, the average yield of α137–141 was around 34%. Considering the maximum production of α137–141 at 3 h, it was estimated that after around only 30 min of hydrolysis (i.e., a DH of 5% is equivalent around to 21%), 63% of the finally generated peptide was already generated. As previously observed, the production of the peptide of interest progressed rapidly, correlating the trend towards the plateau at the end of the reaction. There were no significant differences between the bovine and human hemoglobin peptide hydrolysis.

### 2.5. Peptidomics Approach to Characterizing the Peptide Populations

The peptide heterogeneity after 3 h of pepsic hydrolysis of bovine and human hemoglobin was determined using a peptidomic approach. Triplicate RP-HPLC-MS/MS runs for both hydrolysates were performed, and MS and MS/MS data were recorded. As illustrated Figure 8A, irrespective of the species, the number of MS (around 6000) and MS/MS (around 10,600) scans (compared two-by-two) were not significantly different (*p* < 0.05), indicating that the MS and MS/MS data were comparable between species. Therefore, Peaks^®^ StudioXPro (v 10.6) identified 217 and 189 unique peptides from bovine and human hemoglobin hydrolysates, respectively. Among the latter, only 57 were unique peptides strictly common between species (Figure 8B).

As illustrated in Figure 8C, among the 217 unique peptides identified from the bovine hemoglobin hydrolysate, 135 and 82 unique peptides came from the alpha- and beta-chains, respectively. In the same way, from the 189 unique peptides from humans, 116 and 74 were from the alpha and beta chains (Figure 8D), respectively. Figure 8C,D illustrates, according to the same color code through the 2D and 3D heat maps, the identified-peptide repartition all along of the amino acid backbones for each hemoglobin chain. The higher the occurrence, the more the peptide zone tends towards red then black. Therefore, these heat maps underline the high abundance of peptides detected through the red- to black-colored protein regions, where proteolysis is limited, and the low abundance of detected peptides in the non-colored protein regions, where the proteolysis is important.

Respective to species, the 2D heat maps of alpha-chains (Figure 8C) mainly highlight three to four protein regions (in orange, red and black) where the peptides are resistant to pepsin hydrolysis due to the high number of unique peptides identified in these regions. For the bovine hemoglobin alpha-chain, pepsin-resistant protein regions are sources of bioactive peptides (see Figure 8C, upper panel). Unfortunately, the same comparison cannot be made for the human one due to the absence of bioactive peptides descripted from the human hemoglobin. More interesting, probably due to the high similarity between the 3D structures of the alpha-chains of both species and despite the acid denaturation of 3D protein structures, three pepsin-resistant protein regions were qualitatively identical between species (see 3D structure regions colored either in orange, red, or black), and only quantitative differences related to the number of peptides identified, exist. Note that two of the three pepsin-resistant protein regions surround the heme (depicted in grey color). Concomitantly, only one pepsin-resistant protein region of the bovine hemoglobin alpha-chain (the bottom alpha-helix) is not recovered from the human one.

It is interesting to note that the three-dimensional structure of hemoglobin subunits has been remarkably conserved in all vertebrates since the emergence of the ancestral molecule, which likely dates back around 600 to 700 million years. As a result, the sequence of hemoglobin is also predominantly conserved between these two species.

Inversely, the 2D heat maps of the hemoglobin beta chains (Figure 8D) appear globally distinct, as three pepsin-resistant protein regions (colored in orange) are highlighted from the bovine beta-chain compared with only one for the human one. Through comparison with the 2D heat map of the bioactive peptides (Figure 8D, upper panel) from the bovine beta chain, only the third pepsin-resistant protein region is a source of bioactive peptides. Surprisingly, the 3D heat maps reveal that no matching between the 3D structures of the three pepsin-resistant protein regions of the bovine hemoglobin beta-chain and the human one. In this regard, it should be noted that our search did not find any studies specifically focused on the enzymatic hydrolysis of hemoglobin that already present peptidomics results combined with 3D structures. Although other papers have already been published with such results on other types of proteins [29,33], we suspect that this might be the first time for hemoglobin.

### 2.6. Focus on Bioactive Peptides from Bovine and Human Hemoglobin Hydrolysis

By performing enzymatic hydrolysis using pepsin on bovine and human hemoglobin, a large population of bioactive peptides was produced. Some peptides identified by UPLC-MS/MS displayed antimicrobial activities [11,13,34], hematopoietic activities [10], opioid activities [8,9], bradykinin potentiating activities [8,9], coronary constrictor activities [35], antihypertensive activities [11], antioxidant activities [15], and anticancer activities [36].

Table 1 below presents the bioactive peptide sequences identified by peptidomics after the hydrolysis of bovine and human hemoglobin. These results allowed for the comparison of peptides between the two species and highlighted the similarities or differences in their composition. Indeed, the majority of bioactive peptides identified in the bovine were also present in the human hemoglobin without modifications. However, some antimicrobial peptides present in the bovine hemoglobin were not found as such in the human hemoglobin, probably due to differences in the cleavage sites or modifications of the peptide sequence. For example, the antimicrobial peptide KLLSHSL located at α99–105 became KLLSHCL in the human hemoglobin. Similarly, the growth-stimulating peptide STADA located at β48–52 became STPDA in humans. These observations directly corroborate the results of the bioinformatics approach previously performed. Following these observations, the presence of several new bioactive peptides in human hemoglobin was highlighted. Although some of these peptides were already known in bovine hemoglobin, this study allowed for the discovery of new bioactive peptides in human hemoglobin, such as antibacterial peptides (α37–46, PTTKTYFPHF; α36–45, FPTTKTYFPH; α137–141, TSKYR, α133–141 STVLTSKYR), opioid peptides (α137–141, TSKYR; β31–40, LVVYPWTQRF; β32–40, VVYPWTQRF), an ACE inhibitor (β129–135, KVVAGVA), an anticancer agent (β33–39, VVYPWTQ), and an antioxidant (α137–141, TSKYR). These peptides have never been identified in human hemoglobin before, to the best of our knowledge. These results highlight the potential of human hemoglobin as a source of bioactive peptides useful for the food or pharmaceutical industry.

## 3. Discussion

### 3.1. Reaction Mechanism and Production of the Active Peptide α137–141 during the Enzymatic Hydrolysis of Human and Bovine Hemoglobin

This study confirms the great similarities between bovine and human hemoglobin. In silico analysis of the peptide sequences was used to predict the peptides derived from the peptide hydrolysis of bovine and human hemoglobin, providing information on the potential bioactive peptides generated during digestion. The blast results showed a high identity, 88% and 85%, indicating a high similarity between the α and β chains, respectively. Peptide Cutter software was used to predict the cleavage sites during peptide hydrolysis, revealing major conservation in the number and location of cleavage sites between the two species, while highlighting some differences. Some peptides were conserved, notably our target peptide (TSKYR), while others were specific to each species. Numerous studies have used similar approaches to compare the peptide sequences between different hemoglobin types, showing similarities in conserved peptides as well as specific differences in peptides unique to each species [37,38]. These findings are consistent with our own predictions, confirming the coherence between different studies. These results have provided us with invaluable tools for furthering our understanding of the functional properties and have enabled us to carry out practical biological experiments, in particular enzymatic hydrolysis through pepsin.

The studies carried out on the hydrolysis of human hemoglobin revealed that the enzymatic process involved the same reaction mechanism as that observed in bovine hemoglobin. There are several reasons for this observation. Firstly, the hydrolysis conditions applied (23 °C, pH 3.5) and at CBH; CHH of 1% (*w*/*v*) led to a similar evolution in the degree of hydrolysis, with curves showing the same shape as those previously reported in the literature [15]. Furthermore, the chromatograms obtained for identical hydrolysis times for both bovine and human hemoglobin showed remarkable similarities, indicating the similar accessibility of pepsin to its substrate. This can be attributed to the high similarity between the amino acids of the two species, as previously demonstrated in the in-silico study, with identities of 88% for alpha chains and 85% for beta chains.

The kinetics for obtaining peptides from bovine hemoglobin and from human hemoglobin were subject to the zipper mechanism. Under the experimental conditions used here (23 °C, pH 3.5), purified human hemoglobin was in denatured form [34]. The accessibility of the polypeptide chains making up the subunits of the native protein was thus increased for the enzyme. This favored the so-called zipper mechanism. This enabled all of the peptides to be obtained from the substrate under the given conditions. One mole of enzyme degraded one mole of substrate at a time [20]. As a result, throughout proteolysis, the peptide concentrations increased in a proportional and time-limiting manner. On the other hand, the peptides were all present in a qualitatively identical manner, whatever the duration of hydrolysis. The main advantage of using conditions that enabled a zipper mechanism was the ability to influence the total peptide population, both qualitatively (number and variety of peptides present in the hydrolysate) and quantitatively (relative concentrations of the peptides). Thus, the preliminary control for obtaining several DHs was particularly interesting in the context of a project to separate a given peptide. Indeed, each DH corresponded to a specific population of peptides whose concentrations were relative and variable. For example, it was possible to determine the peptide environments where the separation of a given peptide would be preferential. In short, the zipper mechanism enabled the sub-units of human hemoglobin to be hydrolyzed rapidly into peptides with lower molecular weights. These are known as ‘intermediate peptides. These, in turn, were digested into shorter peptides, until the so-called ‘final’ peptides accumulated throughout the hydrolysis reaction. It has also been previously described that low DHs allowed for the greatest accumulation of peptides with an antimicrobial activity [13,28,34]. These included intermediate peptides with sequences of more than 20 amino acids [39], organized in secondary structures [40].

The same mechanism was observed in other studies of hemoglobin hydrolysis and has been described by Nedjar-Arroume et al. [13,15,32]. Furthermore, it was found that this mechanism remained identical, whatever the initial concentration of bovine/human hemoglobin used.

For the peptide of interest, α137–141, RP-UPLC analyses revealed that peptide identification was unaffected by increasing the concentration of bovine/human hemoglobin. These measurements not only confirmed the production of α137–141 at a higher hemoglobin concentration, but also demonstrated that its production was proportional to the initial hemoglobin concentration. Its production was identical, both qualitatively and quantitatively, in the model bovine hemoglobin and in human hemoglobin. This means that hemoglobin could be processed more quickly in a given timeframe.

It should also be noted that after 3 h, 687.98 ± 75.77 mg·L^−1^ could be produced at an initial concentration of bovine hemoglobin in cruor of 10% (*w*/*v*), a concentration approximately ten times greater than that obtained at 1% (*w*/*v*). In addition, it is particularly remarkable that more than 63% of the total α137–141 peptide production was achieved in only 30 min of hydrolysis, with the peak production being reached at 3 h. This observation could be explained by the presence of new peptides generated during hydrolysis, which became competitive substrates for the enzyme, thus slowing the overall rate of hydrolysis [19,36]. These results have important implications in terms of the feasibility and profitability of a production process targeting a specific peptide. They raise questions about the optimal strategy for maximizing the production of this given peptide in a given time frame.

### 3.2. Application of the Peptidomics Approach to the Study of Peptide Populations

The peptidomics approach was used to characterize peptide heterogeneity after 3 h of peptide hydrolysis of bovine and human hemoglobin. The MS and MS/MS data obtained were comparable between the two species, with a similar number of MS and MS/MS scans. The analysis revealed the identification of 217 unique peptides in bovine hemoglobin hydrolysate and 189 unique peptides in human hemoglobin hydrolysate. Of these, 57 peptides were strictly common to both species. This approach produced a large population of bioactive peptides with antimicrobial [11,32,37], hematopoietic [10], opioid [8,9], bradykinin potentiating [8,11], coronary constrictor [39], antihypertensive [11], antioxidant [15], and anticancer [40] activities. Most of the bioactive peptides identified in bovine hemoglobin were also present in human hemoglobin, without modification. However, some antimicrobial peptides specific to bovine hemoglobin were not found in human hemoglobin, which may be attributed to the differences in cleavage sites or modifications of the peptide sequence, in agreement with the results of the previous bioinformatics analysis.

This study made a significant discovery by identifying novel bioactive peptides in human hemoglobin. These peptides include antibacterial peptides (4), opioid peptides (3), an ACE inhibitor (1), an anticancer agent (1), and an antioxidant (1). This finding is remarkable, because these peptides were already known to be present in bovine hemoglobin, but their presence in human hemoglobin had not been established until now. These results highlight the potential of human hemoglobin as a valuable source of bioactive peptides, opening up new prospects for their use in the food and pharmaceutical industries.

The study focuses on the analysis of antimicrobial peptides derived from the hydrolysis of hemoglobin. However, it is important to note that different peptide populations are obtained at different degrees of hydrolysis (DH) [15]. Generally, a peptide population has a lower molecular weight when the DH is higher. In this study, hydrolysis was carried out for 3 h, corresponding to a DH of 10. To further investigate these observations, a study was conducted on all of the antimicrobial peptides identified during our experiments, as well as those reported in the literature [10,12,31,40,41,42]. The results of this study are presented in Table 2.

Several studies have highlighted the classification of peptides resulting from the hydrolysis of hemoglobin by pepsin into different families [8]. Antimicrobial peptides derived from bovine and human hemoglobin hydrolysis have thus been grouped into four distinct families, three in the α chain and one in the β chain of hemoglobin. This discovery suggests a possible structural and functional similarity between these different families of peptides.

The first family located on the N-terminal side of the α chain of bovine hemoglobin is composed of peptides α1–32, α1–29, α1–28, α1–27, and α1–23. The active part of these peptides is thus located between residues 1 and 23. However, most of these antimicrobial peptides have also been isolated from the hydrolysis of human hemoglobin [41]. Among these pure peptides, peptides α1–40, α1–33, α1–32, α1–31, α1–29, and α1–20 have been identified. In this region of the α chain, there are only four amino acids (AAs) that differ between the two sequences (Figure 9). This shows that the change in four AAs (proline to alanine, threonine to glycine, alanine to glycine, and glycine to alanine) did not result in a total loss of antimicrobial activity.

The second family located between residues 33 and 98 includes peptides α33–98, α33–97, α34–98, α36–97, α37–98, α33–83, α34–83, α33–66, and α34–66 in cattle and α35–56, α35–80, and α35–77 in humans. The following peptides are present in both species without modification: α33–46, α34–46, α36–45, and α37–46. The active sequence is located between residues 36 and 46. Indeed, this active sequence is present in both humans and cattle.

The last family located on the C-terminal side of the α chain is composed, on the one hand of the following peptides: α107–141, α107–136, α107–133 (bovine), α110–131 (human), α137–141, and α133–141 (human and bovine). On the other hand, peptides α99–105, α100–105, and α99–106 are listed in both species, with only one AA that differs.

On the β chain side of bovine and human hemoglobin, peptide β1–13 (bovine), β43–83 (human), and β1–30 (in both species) has been identified, as well as a family of peptides located in the C-terminal region of the β chain. The first peptide forming this family corresponds to peptide β114–145, which then generates peptide β121–145, itself a precursor of peptide β126–145. Similarly, in humans, peptides β111–146 and β115–146 are part of this family, and finally, peptide β140–145 is present in both species. It is then possible to specify the active part within these peptides, which is located between residues 126 and 146.

In summary, the comparative study of the hydrolysis of human and bovine hemoglobin has made it possible to classify the vast majority of human and bovine antimicrobial peptides into the same families due to the presence of common active sites. This suggests a possible structural and functional similarity.

## 4. Materials and Methods

### 4.1. Comparison of Peptide Sequences by Bioinformatics Approach

The software used for this study is the Protein Basic Local Alignment Search Tool (BLAST^®^ version 2.9.0+), which is freely available online at the following site: https://blast.ncbi.nlm.nih.gov/Blast.cgi (consulted on 24 July 2023).

The two proteins studied are Bovine hemoglobin, a model protein of our laboratory, the Charles Violette Institute, BioEcoAgro laboratory, and human hemoglobin.

The algorithm used by blast is the Smith Waterman algorithm, which allows for fast alignment while maintaining a good sensitivity. The reference amino acid sequences were obtained from the following database: https://www.ncbi.nlm.nih.gov/ (consulted on 24 July 2023). 

### 4.2. Prediction of Clipping Sites by Bioinformatics Approach

Peptide Cutter software from ExPASy (Expert Protein Analysis System), was used for the prediction of the peptide population resulting from the pepsin hydrolysis of bovine or human hemoglobin. It is accessible free of charge at following the link: https://web.expasy.org/peptide_cutter/ (consulted on 24 July 2023). A protein sequence generated by proteases can be predicted to contain possible cleavage sites using the Peptide Cutter tool. This software delivers the query sequence along with a table of the positions of potential cleavage sites that can be mapped to it.

### 4.3. Materials: Reagents, Solvents and Standards Used

All of the solvents and chemicals used in this study were of analytical grade and were sourced from commercial suppliers such as Sigma-Aldrich (Saint-Quentin Fallavier, France) or Flandre Chimie (Villeneuve d’Ascq, France). A Milli-Q system was used in the lab to prepare the ultrapure water.

-Purified bovine hemoglobin powder (H2625), dark brown, and purified human hemoglobin (H7379), dark red, were obtained from Sigma-Aldrich (St. Louis, MI, USA). The hemoglobin were stored at 4 °C before use.-Pepsin, a lyophilized powder sourced from porcine gastric mucosa, purchased from Sigma-Aldrich (P6887). Its activity was measured at 3250 AU/mg protein using a protocol established by the supplier. To maintain the stability of the pepsin, it was stored at a −20 °C.

Standard neokyotorphin (α137–141) was purchased from GeneCust (Boynes, France) and stored at −20 °C before use.

### 4.4. Preparation of Hydrolysates

The classical enzymatic hydrolysis using pepsin in a beaker, patented by the laboratory, was used to compare the bovine hemoglobin with human hemoglobin. Bovine hemoglobin was considered as the “control” as numerous enzymatic hydrolysis studies have proven it can be used for the production of active peptides and especially the peptide α137–141 with various biological activities [13,15,23,28]. All of the results were compared to this control.

The α137–141 peptide is a sequence of five amino acids, namely Thr–Ser–Lys–Tyr–Arg (TSKYR), located at the C-terminus of the α-chain of bovine hemoglobin. It has a monoisotopic molar mass of 653.339 Da and an isoelectric point of 10.5. It is thus charged +2 at pH 7 [28].

#### 4.4.1. Preparation of the Stock Solution

A stock solution was prepared by adding 15 g of bovine hemoglobin (BH) or human hemoglobin (HH) in 100 mL of ultrapure water. The supernatant was recovered after centrifugation at 4000 min^−1^ for 30 min (Eppendorf AG, 22,331 Hamburg, Germany; Centrifuge 5804 R, Brinkmann Instruments, Westbury, NY, USA). The Drabkin method established by Crosby, Munn, and Furth (1954) was used to determine the real concentrations of BH (C_BH_) and HH (C_HH_) (1954). This is a spectrophotometric method used to quantitatively determine the hemoglobin concentration in whole blood. For this purpose, 20 µL of the sample was added to 10 mL of Drabkin’s reagent D5941, Sigma-Aldrich, for 15 min at room temperature and it was protected from light. Then, the absorbance was measured using a UV spectrophotometer (ChemStation UV–VIS spectrophotometer 8453A, Agilent Technologies, Santa Clara, CA, USA) at a wavelength of 540 nm. The measurements were reported on the calibration curve. From this concentration C of the bovine or human stock solution, several hemoglobin solutions were prepared by dilution to the following precise concentrations C_BH_ and C_HH_: 1%, 2%, 8% and 10% (*w*/*v*).

#### 4.4.2. Hydrolysis Process

Initially, the stock solutions of human or bovine hemoglobin were in their native, so-called “globular”, form, with two alpha, beta chains, and heme. They were then denatured by lowering the pH to 3.5 by adding 2 M HCl. It is found in an acidic environment in an “expanded” form. This promoted the attack of the alpha and beta chains by the enzyme. The hydrolysis process was carried out at a pH of 3.5 through the gradual addition of porcine pepsin (Sigma-Aldrich, -EC 3.4.23.1, 3200–4500 units mg^−1^ protein) that had been previously dissolved in ultrapure water. An enzyme/substrate ratio of 1/11 (mole/mole) was used. Porcine pepsin is a type of endopeptidase that belongs to the aspartyl protease family. It selectively cleaves hydrophobic or aromatic amino acid residues in peptide bonds, as described by Dunn in 2002 [43]. The samples were taken for the following hydrolysis times: 0, 2.5, 5, 15, 30, 60, 120, and 180 min, corresponding to different degrees of hydrolysis. Then, the hydrolysis reaction was stopped by adding 5 M NaOH until reaching a final pH of 9, which deactivated the enzyme. The pH was measured with a fine tip pH electrode (Mettler Toledo, LE422, micro, 3 mm, Leicester, UK).

The samples were stored at −20 °C to avoid basic hydrolysis. Finally, three copies were made for each species.

### 4.5. Determination of the Degree of Bovine and Human Hemoglobin Hydrolysis

DH, which stands for degree of hydrolysis, is a measure of the ratio of the number of peptide bonds cleaved by an enzyme to the total number of substrate (protein) bonds. It is calculated using Equation (1):(1)$$\mathrm{DH}=100\;\times \frac{h}{\;h0}$$
where DH is expressed as a percentage (%), *h* is the number of hydrolyzed peptide bonds in the substrate, and *h*0 is the total number of peptide bonds in the protein substrate.

DH is quantified here by the ortho-phthaldialdehyde (OPA) technique described by Church et al., 1983 [44] and Spellman et al. (2003) [45]. To determine the number *h* of hydrolyzed bonds, Equation (2) was used:(2)$$h=\frac{\Delta \mathrm{Abs}\cdot M\cdot \mathrm{DF}\;}{\varepsilon \cdot \mathrm{CBH}/\mathrm{HH}}$$
where ∆Abs is the difference between the absorbance at 340 nm of the hydrolyzed sample and the absorbance of the unhydrolyzed sample, M the monoisotopic masses of the protein (for bovine hemoglobin: 64.500 Da and for human hemoglobin: 64.458 Da), DF the dilution factor of the sample, and ε the molar extinction coefficient of the OPA reagent at 340 nm (6000 mol^−1^cm^−1^ according to Church et al.,1985 [46]. C_BH_ and C_HH_ the concentration of bovine/human hemoglobin (10 g·L^−1^ equivalent to 1%).

The reagent was prepared at a minimum of 1 h before use and was stored in the dark. Its composition to obtain a final volume of 50 mL was as follows: 25 mL sodium tetraborate solution (0.1 M) (Fisher Chemicals, Belgium), 5 mL 10% (*w*/*v*) sodium dodecyl sulfate (SDS) (Fisher Scientific in Branchburg, NJ, USA), 10 μL β-mercaptoethanol (Sigma-Aldrich), and 80 mg OPA (Sigma-Aldrich) dissolved in 1 mL methanol (Fisher Chemicals, Brussel, Belgium). The linearity of the reagent was checked by assaying L-leucine (Sigma-Aldrich). To perform the assay, a volume of 100 μL of test sample was added to 2 mL of reagent. After incubating for 2 min at ambient temperature, the absorbance at 340 nm was measured.

### 4.6. Analysis of the Peptide Hydrolysate by RP-UPLC and RP-HPLC

#### 4.6.1. Hardware, Software, and Protocol Used

##### Analysis by RP-UPLC

All of the samples were analyzed by reverse phase ultra-performance liquid chromatography (RP-UPLC) using a Watters ACQUITY UPLC H-Class PLUS (Waters Corporation, Milford, MA, USA) system adjusted to 215 nm.

The hydrolysis fractions were then injected into a C4 Colum (2.6 µm 150 × 2.1 mm) after filtration at 0.22 µm with polyvinylidene difluoride (PVDF) filters. The injection volume was 2.5 μL. The flow rate was set at 0.3 mL·min^−1^.

The elution program was as follows: The mobile phases were ultrapure water/trifluoroacetic acid (99:1, *v*/*v*) as solvent A and acetonitrile/trifluoroacetic acid (99:1, *v*/*v*) as solvent B. A gradient was applied with solvent B increasing from 5% to 30% in 30 min, then to 60% for 10 min and held until 47min at 95%, then back to initial conditions. UV absorbance scans were performed between 200 and 390 nm at a rate of one spectrum per second with a resolution of 4.8 nm. Chromatographic acquisition and analysis were performed with Empower 3 software (Version 3 Waters). Each sample analysis was performed in triplicate to ensure technical reproducibility.

#### 4.6.2. Identification of the α137–141Peptide

Using the peak purity evaluation and spectral comparison, as described previously for the identification of hemorphins and α137–141 peptide, neokyotorphin, in bovine cruor [14,15,32], an initial rapid identification of peptides from bovine and human hemoglobin peptic hydrolysate was carried out through a comparison of the UV spectra. The spectrum of the standard peptide was stored in the digital library of the software. Two other mathematical analyses were then performed. Firstly, a comparison of the UV spectra between the standard peptide and the peptide presents in the peak was made and then the purity of the peak was estimated.

For the spectral comparison, two parameters were studied. The first that was considered was the match angle (MA), or “angle of encounter”. The latter represented the measure of the difference in spectral patterns between the standard peptide and the peptide to be identified. MA ranged from 0 to 90 degrees. The smaller this value, the more similar the spectra were considered to be. Conversely, values approaching 90 degrees indicated large differences between the peptides from a spectral perspective.

The second measure was the match threshold (MT) parameter. This indicated the sensitivity of the measurement method, allowing for quantification by an angle that ranged from 0 to 180 degrees. The larger the value, the more sensitive the method was found to be. However, if the MA was greater than MT, it could show that the two spectra were different. On the other hand, if MA was lower than MT, there was a good chance that the two spectra being compared were similar or even the same [32]. Therefore, identical spectra could identify expected peptides.

A second measurement was performed to estimate the purity of a peak from the chromatographic profile in addition to the spectral comparison.

To do this, two criteria were expressed, the Purity Angle (PA), represented the relative spectral homogeneity within a peak. This value could range from 0 to 90 degrees, with 0 indicating perfect spectral homogeneity. Next, the purity threshold (PT) included a background, high sample concentration, photometric error, and/or solvent. This could result in a higher PA than what it actually was.

At the end, if PA was lower than PT, the measured spectra could be considered similar considering the inherent interferences of the method. However, this did not prove chemical purity.

For each measurement, the lowest values foreach of the parameters were sought, respecting MA being lower than MT and PA being lower than PT. In this way, the α137–141 assay was the most realistic and reproducible between the different samples while avoiding the inclusion of other compounds or peptides in the measurements [30,34].

#### 4.6.3. Quantification of α137–141 in Hydrolysates

A standard range was performed by injecting the standard α137–141 peptide at concentrations ranging from 0 to 5 mg·mL^−1^. The areas under the peaks were extracted from the chromatographic profiles at 215 nm. *A* linear relationship (r^2^ = 1) was established between the concentration of α137–141 and the peak area of the peptide of interest within the hydrolysates, using the following equation:*C*∝137–141 = 4419.6 × *A*∝137–141
where *C*α137–141 is the concentration of α137–141 present in the sample (expressed in mg·L^−1^) and *A*α137–141 as the peak area (μV·s). In addition, the injection of the standard allowed for the identification of α137–141 at a retention time of approximately 5 min under the elution conditions described previously.

### 4.7. Peptidomics Approach

After centrifugation for 10 min at 8000× *g*, aliquots of 10 µL of human and bovine hydrolysate at a concentration of 30 mg·mL^−^^1^ were analyzed in triplicate by RP-HPLC-MS/MS, (ACQUITY UPLC system; Waters Corporation, France). Peptides were separated using a C18 column (150 × 3.0 mm, 2.6 µm, Uptisphere CS EVOLUTION, Interchim, France). The mobile phases consisted of solvent A (0.1% (*v*/*v*), formic acid/99.9% (*v*/*v*), and water) and solvent B (0.1% (*v*/*v*), formic acid/99.9% (*v*/*v*), and acetonitrile (ACN)). The ACN gradient (flow rate 0.5 mL·min^−^^1^) was as follows: from 5% to 30% solvent B over 40 min, from 30% to 100% solvent B over 10 min, followed by washing and equilibrating procedures using successively 100% and 1% of solvent B for 5 min each. The eluate was directed into the electrospray ionization source of the qTOFSynapt G2-Si™ (Waters Corporation, Manchester, UK), previously calibrated using a sodium format solution. Mass spectrometry (MS) measures were performed in sensitivity, positive ion, and data dependent analysis (DDA) modes using the proprietary Mass Lynx 4.2 software (Waters). The source temperature was set at 150 °C and the capillary and cone voltages were set to 3000 and 60 V·MS. The data were collected for *m*/*z* values in the range of 50 and 2000 Da with a scan time of 0.2 s. A maximum of 10 precursor ions were chosen for the MS/MS analysis with an intensity threshold of 10,000. MS/MS data were collected under the collision-induced dissociation (CID) fragmentation mode and a scan time of 0.1 with specified voltages ranging from 8 to 9 V and from 40 to 90 V for the lower and higher molecular mass ions, respectively. The leucin and enkephalin ([M + H]^+^ of 556.632) was injected in the system every 2 min for 0.5 s to follow and correct the measure error during the entire time of analysis.

Protein database searches to identify the chromatographed peptides were performed in the UniProt data bases restricted to *Bos taurus* and *Homo sapiens* (accessed online in July 2022) via PEAKS^®^ Studio XPro (v 10.6) (Bioinformatics Solutions Inc., Waterloo, ON, Canada). A mass tolerance of 35 ppm and an MS/MS tolerance of 0.2 Da were allowed. The data searches were performed with notifying pepsin as the hydrolysis enzyme. The relevance of the protein and peptide identities was judged according to their identification score in the research software (using a *p*-value < 0.05 and a false discovery rate < 1%).

### 4.8. Two-Dimensional (2D) and Tri-Dimensional (3D) Heatmaps

For each hydrolysate, the peptide abundance all along of the amino acid sequences was assessed using a home-built Microsoft Excel heat map giving the amino acid occurrences in the proteins (from the N-terminal amino acid (left) to the C-terminal amino acid (right). For each amino acid of the protein sequence, a score was calculated according to its occurrence in each unique peptide sequence identified. The higher the occurrence, the more the peptide zone tended towards red then black. The 3D structures of bovine and human hemoglobin alpha- and beta-chains were uploaded (access on March 2023) from RSCB Protein Databank (PDB) using the SWISS-MODEL bioinformatics web-server (PDBID were 1hda.1.C, 6ihx.1.D, 1bz1.1.A, 6kye.1.D for bovine alpha- and beta-chains and human alpha and beta chains, respectively).

### 4.9. RP-UPLC Statistical Analysis

Three independent replications of each condition were run, and each analysis was carried out in triplicate. One-way or two-way analyses of variance were performed on the data (ANOVA). GraphPad Prism 8.0.2 (263) was used to conduct multiple *t* test comparison tests on the data to identify the treatment that was statistically different from the others at a probability level of 0.05 (*p* < 0.05).

## 5. Conclusions

In this comparative study, the hydrolysis of human hemoglobin was explored in parallel with bovine hemoglobin, which is reputed to be rich in bioactive peptides. The focus was on obtaining the α137–141 peptide, known for its antimicrobial potential and natural antioxidant properties, making it highly promising for the food pharmaceutical industry. Bioinformatics alignment was used to compare the human hemoglobin hydrolysate sequences in silico with the bovine hemoglobin hydrolysate. This confirmed the comparability of the two proteins, encouraging research into the target peptide. The results of this study demonstrated that the hydrolysis of human hemoglobin at high initial concentrations (1–10% (*w*/*v*)) resulted in the rapid production of antimicrobial peptide α137–141. The maximum concentration of this final peptide reached 687.98 ± 75.77 mg·L^−1^ in 3 h, with more than 60% produced in just 30 min. This hydrolysis process followed the same enzymatic mechanism as that observed for bovine hemoglobin, confirming the rapidity and efficiency of α137–141 peptide production.

The identification of bioactive peptides by peptidomics revealed that most of the active peptides (particularly antimicrobial peptides) identified in cattle were also present in humans without modification or they belonged to the same family, due to the presence of common active sites. This study also revealed the presence of new bioactive peptides in human hemoglobin, including antibacterial peptides (4), opioid peptides (3), an ACE inhibitor (1), an anticancer agent (1), and an antioxidant (1). Some of these peptides were already known, but others had never been reported before in human hemoglobin.

These encouraging results pave the way for numerous research prospects. Future studies could focus on optimizing the hydrolysis conditions of human hemoglobin in order to maximize the production of the bioactive peptides of interest. This could involve adjusting parameters such as pH, temperature, enzyme concentration, and hydrolysis time to obtain even higher yields. In addition, it would be relevant to carry out functional studies on bioactive peptides in in vitro and in vivo models. These studies could include tests for antimicrobial activity, antioxidant effect, or specific pharmacological activity, making it possible to demonstrate their efficacy and safety of use in relevant biological contexts.

## Figures and Tables

**Figure 1 ijms-24-11921-f001:**
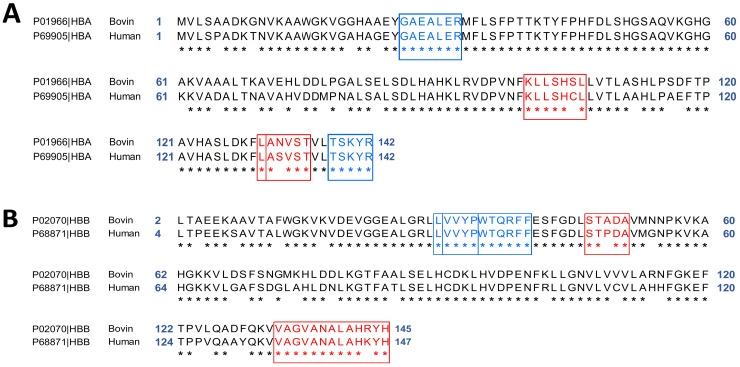
Blast of the hemoglobin alpha chain (**A**) and beta chain (**B**) from *Bos taurus* (bovine hemoglobin) and *Homo sapiens* (human hemoglobin). (**A**) Conservation of bioactive peptides GAEALER (hematopoietic) and TSKYR in blue (opioid, antioxidant, and antimicrobial). Loss of peptides LANVST (Analgesic and Bradykinin Potentiator), ANVST (DPP-IV Inhibitor), and KLLSHSL (Antihypertensive) in red. (**B**) Conservation of the bioactive peptides VVYP (lipid-lowering), LVVYPWTQRFF (opioid) in blue and loss of the peptides VAGVANALAHKYR (coronary constrictor) and STADA (bacterial growth stimulator) in red. Blasts were performed using Local Similarity Program (SIM) from ExPASy. *: mean that the amino acids are exactly the same in the alpha (**A**) and beta (**B**) chains of the hemoglobins of *Bos taurus* and *Homo sapiens*.

**Figure 2 ijms-24-11921-f002:**
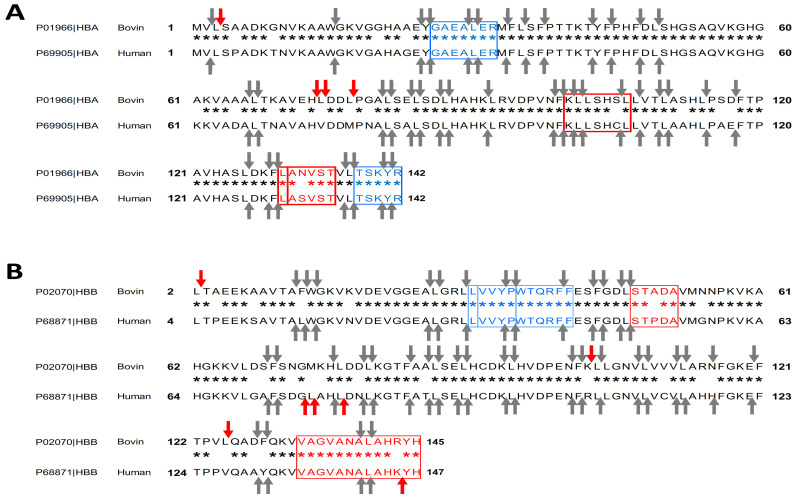
Peptide Cutter used for the potential cleavage sites of the alpha (**A**) and beta (**B**) chains of bovine *(Bos taurus)* and human *(Homo sapiens)* hemoglobin by pepsin at pH > 2. The black arrows indicate the same predicted cleavage sites and those in red differ only for one of the two hemoglobin chains. Blasts were performed using Local Similarity Program (SIM) from ExPASy and the cleavage sites were predicted using the Peptide Cutter from ExPASy. *: mean that the amino acids are exactly the same in the alpha (A) and beta (B) chains of the hemoglobins of *Bos taurus* and *Homo sapiens*.

**Figure 3 ijms-24-11921-f003:**
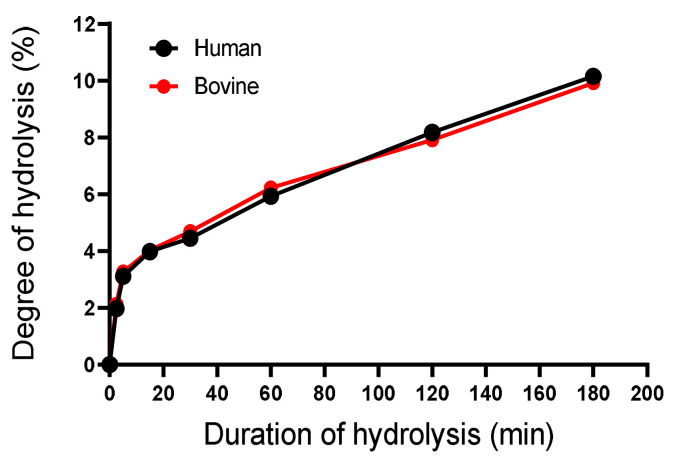
Evolution of the degree of hydrolysis of bovine and human hemoglobin during its hydrolysis treated with porcine pepsin for 180 min (3 h) (pH 3.5, 23 °C, E/S = 1/11, C_BH_/C_HH_ = 1%, *w*/*v*).

**Figure 4 ijms-24-11921-f004:**
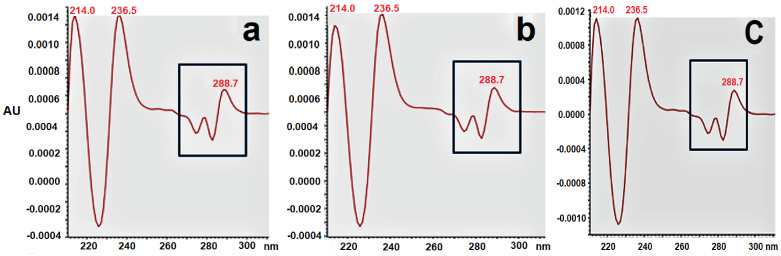
Comparison of the second derivative spectra between the (**a**) standard α137–141 peptide and (**b**) obtained α137–141 peptide from bovine hemoglobin hydrolysis and (**c**) the obtained α137–141 peptide from the human hemoglobin hydrolysis.

**Figure 5 ijms-24-11921-f005:**
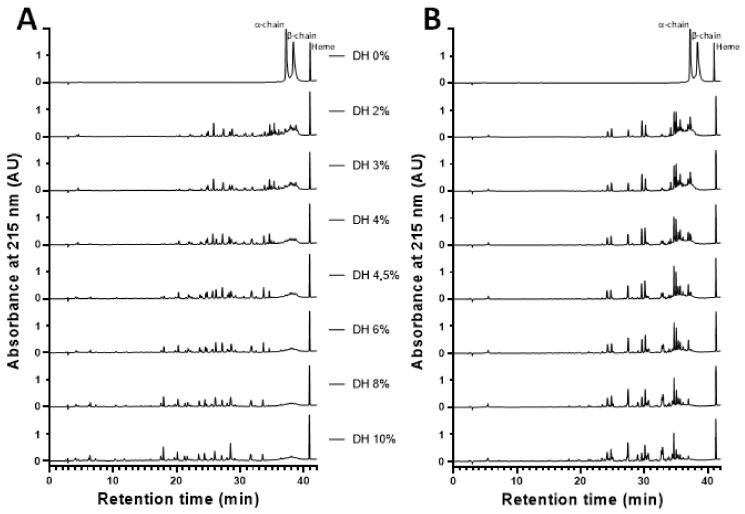
Chromatographic hydrolysis profiles of bovine hemoglobin compared with human hemoglobin at 215 nm using RP-UPLC, analyzed by the C4 type column (pH 3.5, 23 °C, E/S = 1/11, C_BH_ = 1%, C_HH_ = 1%, *w*/*v*) (**A**) bovine hemoglobin and (**B**) human hemoglobin.

**Figure 6 ijms-24-11921-f006:**
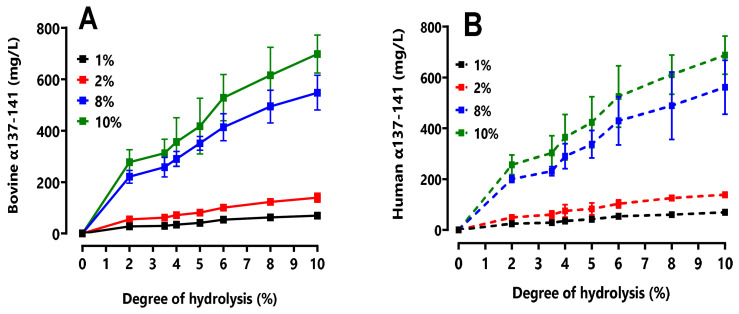
Appearance kinetics of the α137–141 peptide during peptic hydrolysis of bovine hemoglobin BH (**A**) and human hemoglobin HH (**B**) hydrolysis at different bovine (C_BH_) and human (C_HH_) hemoglobin concentrations (*w*/*v*), (pH 3.5, 23 °C, E/S = 1/11). A comparative study was performed between the different graphs in the same condition and no significant difference (ns) was found between the human hemoglobin and bovine hemoglobin (multiple *t*-test comparisons).

**Figure 7 ijms-24-11921-f007:**
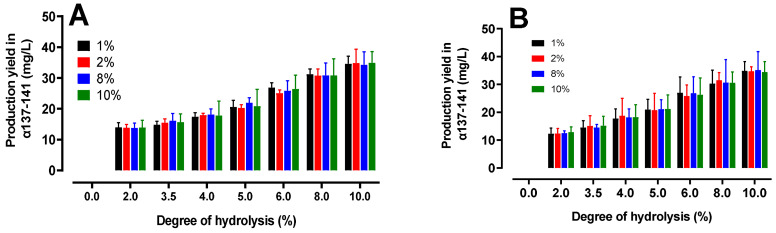
Production yields of α137–141 during the peptic hydrolysis of bovine (**A**) and human (**B**) hemoglobin at different concentrations (*w*/*v*) (pH 3.5, 23 °C, E/S = 1/11). A comparative study was performed between the two histograms and no significant difference (ns) between human hemoglobin and bovine hemoglobin was found (multiple *t*-test comparisons).

**Figure 8 ijms-24-11921-f008:**
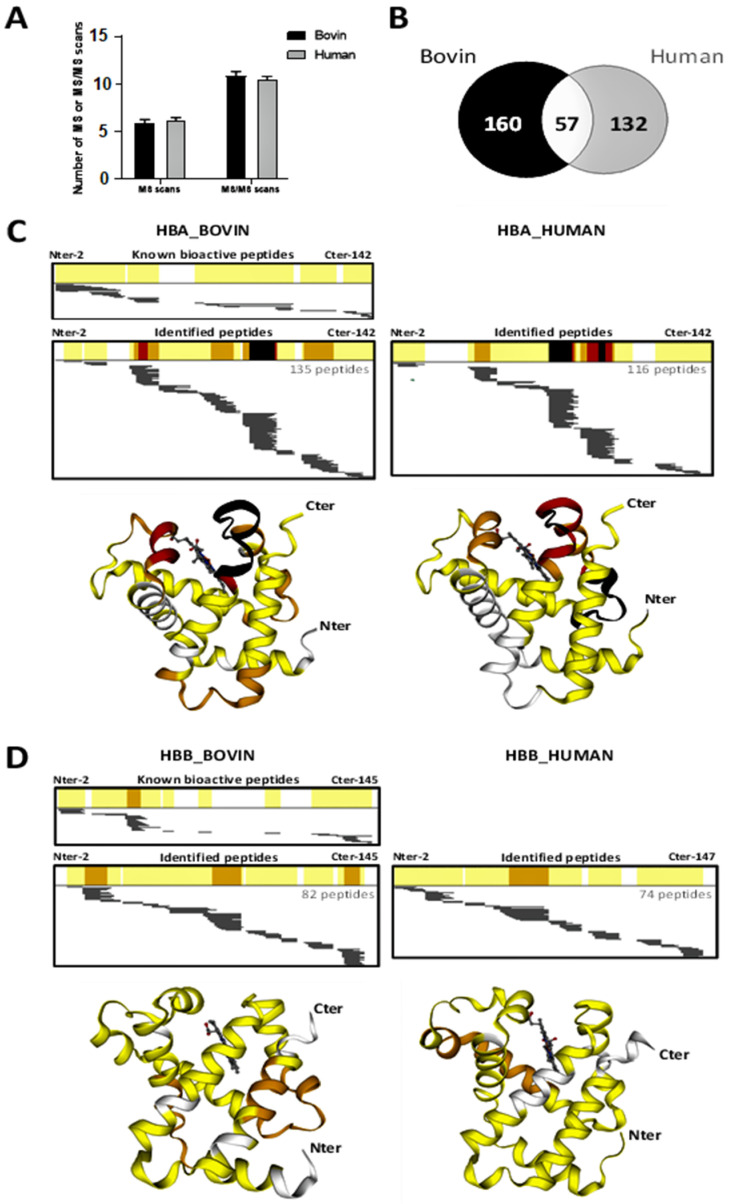
Peptidomics analysis, performed using RP-HPLC-MS/MS and bioinformatics, from bovine and human hemoglobin after a 3h hydrolysis period. (**A**) Number of MS and MS/MS scans. (**B**) Venn diagram showing the number of distinct and common hemoglobin peptides identified between the two hydrolysates. (**C**,**D**) 2D and 3D heat maps highlighting the occurrence of detected peptides along the amino acid sequences for hemoglobin subunit alpha (**C**) and subunit beta (**D**). The color code, corresponding to the peptides containing the amino acids that were identified, is as follows: white color, no peptides containing these amino acid shave been identified; yellow color, between 1 to 10 peptides have been identified; orange color, between 11 to 20 peptides have been identified; red color, between 21 to 30 peptides have been identified; black color, more than 31 peptides containing these amino acids have been identified. The grey bars under the 2D heat maps correspond to the unique peptide sequences identified. Heat maps using the 3D modeling were performed with the SWISS-MODEL bioinformatics web-server. Heme is depicted in grey. The hydrolysates were analyzed in triplicate. N-terminal extremity (Nter) and C-terminal extremity (Cter).

**Figure 9 ijms-24-11921-f009:**
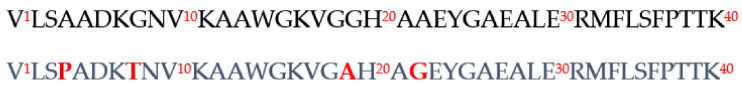
Amino acid sequence of the 1–40 region of the α chain of bovine (black) and human (blue) hemoglobin. The red color represents the variable amino acids.

**Table 1 ijms-24-11921-t001:** Bioactive peptide sequences, identified by peptidomics, resulting from the hydrolysis of bovine and human hemoglobin.

Biological Activity	Position	Sequence	Monoisotopic Molecular (Da)	Bovine Hemoglobin	Human Hemoglobin
Antimicrobial	α34–46	LSFPTTKTYFPHF	1584.787	+	−
	α33–46	FLSFPTTKTYFPHF	1731.855	+	−
	α37–46	PTTKTYFPHF	1237.602	+	+
	α36–45	FPTTKTYFPH	1237.602	+	+
	α137–141	TSKYR	653.339	+	+
	α133–141	STVLTSKYR	1053.571	+	+
	α99–105	KLLSHSL	796.470	+	KLLSH**C**L
	α100–105	LLSHSL	668.375	+	−
	α99–106	KLLSHSLL	909.554	+	KLLSH**C**LL
Hematopoietic	α76–82	LPGALSE	685.354	+	**M**P**N**ALSA
	β115–122	RNFGKEFT	997.487	+	**HH**FGKEFT
Opioid	α137–141	TSKYR	653.339	+	+
	β32–40	VVYPWTQRF	1194.608	+	+
	β31–40	LVVYPWTQRF	1307.692	+	+
	β31–37	LVVYPWT	876.464	+	−
Analgesic and Potentiator of bradykinin	α129–134	LANVST	603.312	+	LA**S**VST
	β129–134	QKVVAG	600.349	+	−
dipeptidyl-peptidase Inhibitor	α130–134	ANVST	490.228	+	A**S**VST
	β6–10	KAAVT	488.285	+	K**S**AVT
ACE inhibition	β129–135	KVVAGVA	642.395	+	+
Antihypertensive	α99–105	KLLSHSL	796.470	+	KLLSH**C**L
Antioxidant	α137–141	TSKYR	653.339	+	+
Bacterial growth stimulator	β48–52	STADA	463.180	+	ST**P**DA
Anticancer	β33–39	VVYPWTQ	891.438	+	+

**Table 2 ijms-24-11921-t002:** Antimicrobial peptides sequences identified by UPLC-MS/MS resulting from the hydrolysis of bovine and human hemoglobin.

Position	Sequence	Monoisotopic Molecular (Da)	Bovine Hemoglobin	Human Hemoglobin	Ref
The first family is situated on the N-terminal end of the α-chain, with the active portion found between residues 1 and 23.					
α1–40	VLSPADKTNVKAAWGKVGAHAGEYGAEALERMFLSFPTTK	4249	−	+	b
α1–33	VLSPADKTNVKAAWGKVGAHAGEYGAEALERMF	3474	−	+	b
α1–32	VLSPADKTNVKAAWGKVGAHAGEYGAEALERM	3327	+	+	a
	VLSAADKGNVKAAWGKVGGHAAEYGAEALERM (bovine)	3257	+	−	
α1–31	VLSPADKTNVKAAWGKVGAHAGEYGAEALER	3196	−	+	b
α1–29	VLSPADKTNVKAAWGKVGAHAGEYGAEAL	2911	−	+	b
	VLSAADKGNVKAAWGKVGGHAAEYGAEAL (bovine)	2841	+	−	a
α1–28	VLSAADKGNVKAAWGKVGGHAAEYGAEA	2728	+	−	a
α1–27	VLSAADKGNVKAAWGKVGGHAAEYGAE	2656	+	−	a
α1–23	VLSAADKGNVKAAWGKVGGHAAE	2237	+	+	a
α1–20	VLSPADKTNVKAAWGKVGAH	2949	−	+	b
α18–44	VGAHAGEYGAEALERMFLSFPTTKTYF	2994	−	+	b
A second family of peptides located between residues 32 and 98, with an active sequence between residues 36 and 46.					
α32–41	FLSFPTTKTY	1204	−	+	c
α33–46	FLSFPTTKTYFPHF	1731	+	+	a/e
α34–46	LSFPTTKTYFPHF	1585	+	+	a/e
α36–45	FPTTKTYFPH	1238	+	+	a/e
α37–46	PTTKTYFPHF	1238	+	+	a/e
α33–98	FLSFPTTKTYFPHFDLSHGSAQVKGHGAKVAAALTKAVEHLDDLPGALSELSDLHAHKLRVDPVNF	7151	+	−	a
α33–97	FLSFPTTKTYFPHFDLSHGSAQVKGHGAKVAAALTKAVEHLDDLPGALSELSDLHAHKLRVDPVN	7004	+	−	a
α34–98	LSFPTTKTYFPHFDLSHGSAQVKGHGAKVAAALTKAVEHLDDLPGALSELSDLHAHKLRVDPVNF	7004	+	−	a
α36–97	SFPTTKTYFPHFDLSHGSAQVKGHGAKVAAALTKAVEHLDDLPGALSELSDLHAHKLRVDPVN	6744	+	−	a
α37–98	PTTKTYFPHFDLSHGSAQVKGHGAKVAAALTKAVEHLDDLPGALSELSDLHAHKLRVDPVNF	6657	+	−	a
α33–83	FLSFPTTKTYFPHFDLSHGSAQVKGHGAKVAAALTKAVEHLDDLPGALSEL	5422	+	−	a
α34–83	LSFPTTKTYFPHFDLSHGSAQVKGHGAKVAAALTKAVEHLDDLPGALSEL	5274	+	−	a
α33–66	FLSFPTTKTYFPHFDLSHGSAQVKGHGAKVAAAL	3632	+	−	a
α34–66	LSFPTTKTYFPHFDLSHGSAQVKGHGAKVAAAL	3484	+	−	a
α35–56	SFPTTKTYFPHFDLSHGSAQVK	2495	−	+	b
α35–80	SFPTTKTYFPHFDLSHGSAQVKGHGKKVADALTNAVAHVDDMPNAL	4922	−	+	b
α35–77	SFPTTKTYFPHFDLSHGSAQVKGHGKKVADALTNAVAHVDMP	4624	−	+	b
The third family is located on the c-terminal side of the α-chain.					
α110–131	AAHLPAEFTPAVHASLDKFLAS	2293	+	−	a
α107–141	VTLASHLPSDFTPAVHASLDKFLANVSTVLTSKYR	3788	+	−	a
α107–136	VTLASHLPSDFTPAVHASLDKFLANVSTVL	3152	+	−	a
α107–133	VTLASHLPSDFTPAVHASLDKFLANVS	2838	+	−	a
α133–141	STVLTSKYR	1055	+	+	a/e
α137–141	TSKYR	654	+	+	a/e
α99–105	KLLSHSL (bovine)	796	+	−	a
	KLLSHCL	813	−	+	e
α100–105	LLSHSL	668	+	−	a
α99–106	KLLSHSLL	910	+	−	a
	KLLSHCLL	926	−	+	e
The last family of peptides is located in the c terminal region of the β chain of hemoglobin					
β1–30	MLTAEEKAAVTAFWGKVKVDEVGGEALGRL (bovine)	3176	+	+	a
	MVH LTPEEKSA VTALWGKVNVDEVGGEALG	3137			
β1–55	MVHLTPEEKSAVTALWGKVNVDEVGGEALGRLLVVYPWTQRFFESFGDLSTPDAV	6063	−	+	d
β56–146	MGNPKVKAHGKKVLGAFSDGLAHLDNLKGTFATLSELHCDKLHVDPENFRLLGNVLVCVLAHHFGKEFTPPVQAAYQKVVAGVANALAHKY	9815	−	+	d
β116–146	LLGNVLVCVLAHHFGKEFTPPVQAAYQKVVAGVANALAHKY	4375	−	+	d
β56–72	MGNPKVKAHGKKVLGAF	1782	−	+	d
β43–83	RFFESFGDLSTPDAVMGNPKVKAHGKKVLGAFSDGLAHLDNLK	4616	−	+	b
β111–146	LVCVLAHHFGKEFTPPVQAAYQKVVAGVANALAHKY	3878	−	+	b
β115–146	LAHHFGKEFTPPVQAAYQKVVAGVANALAHKY	3463	−	+	b
β114–145	ARNFGKEFTPVLQADFQKVVAGVANALAHRYH	3556	+	−	a
β121–145	FTPVLQADFQKVVAGVANALAHRYH	2753	−	+	b
β126–145	QADFQKVVAGVANALAHRYH	2196	+	−	b
β1–13	MLTAEEKAAVTAF	1381	+	−	a
β140–145	LAHRYH	795	+	−	a

Ref a [11,13,34], b [41]; c [42]; d [42]; e (The peptides are identified during our UPLC-MS/MS analyses).

## Data Availability

Not applicable.

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
