# Peer review of "Potential of Human Hemoglobin as a Source of Bioactive Peptides: Comparative Study of Enzymatic Hydrolysis with Bovine Hemoglobin and the Production of Active Peptide α137–141"

_ijms, 2023, doi:10.3390/ijms241511921_

Round 1
Reviewer 1 Report
The manuscript described an interesting study on potential of human hemoglobin as a source of bioactive peptides. However, the following problems may limit the academic values of the present study
1) the authors claimed using human hemoglobin as a source of bioactive peptides for the food or pharmaceutical industry. however, human hemoglobin comes from human blood. Is that a serious ethical issue?
2) the authors only focused on the analyses of peptides from enzymatic hydrolysis of hemoglobin. But not separate these peptides and show how much of these peptides they can get from 1 gram hemoglobin and how much hemoglobin they can get from 1 liter human blood
3) the author claimed the bioactivities of these peptides. However, it seemed that they did not do any important bioactivity test or assays to evaluate these peptides by themselves.
Author Response
Reviewer 1
Comments and Suggestions for Authors
The manuscript described an interesting study on potential of human hemoglobin as a source of bioactive peptides. However, the following problems may limit the academic values of the present study.
1.. the authors claimed using human hemoglobin as a source of bioactive peptides for the food or pharmaceutical industry. however, human hemoglobin comes from human blood. Is that a serious ethical issue?
Answer: The supply of human hemoglobin from scientific research organizations, such as the Sigma Aldrich mentioned in the manuscript, can provide assurance as to the ethical issue of samples. These organizations are required to adhere to strict ethical and quality standards to ensure the legality and legitimacy of their products.
We who had used human hemoglobin in the laboratory had the responsibility to ensure transparency and ethics throughout the process (handling notebook). This includes traceability of samples, documentation of their provenance and adherence to established ethical protocols to maintain the trust of relevant stakeholders. Our goal is not to industrialize the use of human hemoglobin, but to find potential active ingredients (peptides). These active ingredients may also be present in bovine haemoglobin which may be the industrial source of these active ingredients. There is the possibility of synthesizing active peptides.
2.. The authors only focused on the analyses of peptides from enzymatic hydrolysis of hemoglobin. But not separate these peptides and show how much of these peptides they can get from 1 gram hemoglobin and how much hemoglobin they can get from 1 liter human blood.
Answer: Hemoglobin was purchased from Sigma-Aldrich as purified haemoglobin powder (bovine ref: H2625 and human: H7379). We hadn't worked on blood.
It should be noted that generally, theblood is essentially subdivided into two parts after centrifugation. The colorless part, plasma, accounts for about 60%. The second part accounts for up to 40% of the blood volume and is responsible for the red color of the blood. It is called "cruor" and contains mainly hemoglobin. About 90% of the mass of human hemoglobin is contained in red blood cells.
By extrapolating concentration data obtained from 1.5 g hemoglobin, where the concentration of α137-141 peptide was 687.98 ± 75.77 mg. L-1, our target peptide, it can be estimated that one gram of hemoglobin would produce about 458.65 ± 50.51 mg. L-1 peptide. With 1.5g representing 10% of 15 g of hemoglobin dissolved in 100 ml of water before the start of the hydrolysis reaction (described in detail in the material and method part).
These calculations are based on the fact that the relationship between the amount of hemoglobin used and the peptide concentration remains linear and proportional. It should be noted that these estimates are only valid for the experimental conditions of our study. Because they can vary according to the specific conditions of the experiment and the factors influencing the enzymatic hydrolysis of hemoglobin by pepsin.
3.. The author claimed the bioactivities of these peptides. However, it appears that they have not conducted bioactivity tests or significant assays to evaluate these peptides on their own.
Answer: Mass spectrometry analysis after 3 hours of enzymatic pepsin hydrolysis identified a list of peptides from both bovine and human hemoglobin. Peptides derived from the hydrolysis of bovine hemoglobin are well documented and some of these peptides, which are claimed to be novel in human hemoglobin, are actually identical without any amino acid modification. Therefore, we can conclude that these peptides share the same activities as those already studied in cattle. For example, our target peptide, TSKYR, was found identically in human hemoglobin. This peptide has already demonstrated antimicrobial and antioxidant activities by our team.
Reviewer 2 Report
This is a study about the hydrolysis products of human hemoglobin and a comparison between human and bovine hemoglobin. This study is valuable for bio-processing for yielding useful peptide products. I have some suggestions helpful for improving the manuscript's quality.
1. In terms of LC-coupled detectors, the authors employed both photo spectrometry and mass spectrometry to detect peptides. However, the chromatography conditions for the two detectors are not consistent. For instance, different reverse phase columns, namely C4 and C18 columns, are respectively used for chromatography with photo spectrometer and mass spectrometer (Peptidomics). It is hard to establish a correspondence between the peptide sequences identified by the mass spectrometer (Table 2) and the fractions measured by the photo spectrometer (Figure 6). The authors should explicitly address this issue in the manuscript and consider resolving it in future studies.
2. It is suggested to upload the MS files with raw data to the PRIDE database (https://www.ebi.ac.uk/pride/) and provide a PRIDE ID in the manuscript.
3. The "Zipper" theory is described unclearly (line 247), and the cited literature is quite old (ref. 21). To help readers understand more directly, it is important to note that, unlike trypsin, pepsin is not a specific-cutting proteinase. Therefore, the production of protein fragments is sensitively influenced by reaction conditions, particularly the pH and temperature, which can cause variations in the resulting products.
4. The term "Degree of Hydrolysis" (DH) is mentioned on line 193, but the abbreviation appears earlier (line 190). Please carefully review and correct the order.
5. Please provide the name of "Waters software" mentioned (lines 204-205), rather than expecting readers to refer to past literature (ref. 16).
5. Please provide the URL link of the mentioned Expasy tool (https://www.expasy.org/).
Author Response
Reviewer 2
Comments and suggestions for authors
This is a study about the hydrolysis products of human hemoglobin and a comparison between human and bovine hemoglobin. This study is valuable for bio-processing for yielding useful peptide products. I have some suggestions helpful for improving the manuscript's quality.
- In terms of LC-coupled detectors, the authors employed both photo spectrometry and mass spectrometry to detect peptides. However, the chromatography conditions for the two detectors are not consistent. For instance, different reverse phase columns, namely C4 and C18 columns, are respectively used for chromatography with photo spectrometer and mass spectrometer (Peptidomics). It is hard to establish a correspondence between the peptide sequences identified by the mass spectrometer (Table 2) and the fractions measured by the photo spectrometer (Figure 6). The authors should explicitly address this issue in the manuscript and consider resolving it in future studies.
Answer: We thank you for this advice and we will make good use of it in our future studies as you have suggested.
- It is suggested to upload the MS files with the raw data to the PRIDE database (https://www.ebi.ac.uk/pride/) and provide a PRIDE identifier in the manuscript..
Answer. We are unable to upload MS files to the PRIDE database. We are willing to send it to you to do so if necessary.
- The "Zipper" theory is described unclearly (line 247), and the cited literature is quite old (ref. 21). To help readers understand more directly, it is important to note that, unlike trypsin, pepsin is not a specific-cutting proteinase. Therefore, the production of protein fragments is sensitively influenced by reaction conditions, particularly the pH and temperature, which can cause variations in the resulting products.
Answer: We had taken note of this advice and will take it into account from now on.
- The term "Degree of Hydrolysis" (DH) is mentioned on line 193, but the abbreviation appears earlier (line 190). Please carefully review and correct the order.
Answer: Corrections made
- Please provide the name of "Waters software" mentioned (lines 204-205), rather than expecting readers to refer to past literature (ref. 16).
Answer: Amendments made.
- Please provide the URL link of the mentioned Expasy tool (https://www.expasy.org/).
Answer: Amendments made.
Reviewer 3 Report
ijms-2491502
Potential of human hemoglobin as a source of bioactive peptides: comparative study of enzymatic hydrolysis with bovine hemoglobin and production of active peptide α137-141
The manuscript describes the structure and mechanism of enzymatic hydrolysis of bovine and human hemoglobin and the potential possibilities for the use of bovine hemoglobin as a source of bioactive peptides (in particular active peptide α137-141).
The study was conducted to evaluate the potential of human hemoglobin as a source of bioactive peptides, compared to bovine hemoglobin, which has been extensively studied in recent years. Specifically, the study focused on neokyotorphin, a natural antimicrobial peptide and meat preservative that corresponds to the α137-141 fragment of bovine hemoglobin obtained by enzymatic hydrolysis.
In the study, the structure of the two types of hemoglobin (human and bovine), the mechanism of enzymatic hydrolysis and the peptides that are the product of this hydrolysis were compared.
Based on the results obtained, this study shows that the two types of hemoglobin have a high similarity for the α and β chains (identities of 88% and 91%, respectively). Also, as a result of subjecting both types of hemoglobin to enzymatic hydrolysis under similar conditions, the hydrolysis of human hemoglobin was found to follow the same reaction mechanism as the hydrolysis of bovine hemoglobin, namely the "zipper" mechanism. The results of mass spectrometric analysis revealed the presence of several bioactive peptides in cattle and humans. Although some were previously known, new bioactive peptides have been discovered in human hemoglobin, such as four antibacterial peptides (α37-41 PTTKTYFPHF; α36-45 FPTTKTYFPH; α137-141 TSKYR; α133-141 STVLTSKYR), four opioid peptides (α137-141 TSKYR; β31-40 LVVYPWTQRF; β31-37 LVVYPWT), ACE inhibitor (β129-135 KVVAGVA), anticancer agent (β33-39 VVYPWTQ) and antioxidant (α137-141 TSKYR). These peptides have never been found in human hemoglobin before, to our knowledge.
In the first part, called "Summary", the authors explain in detail the composition of blood in cattle and man, and also its insufficiently efficient use for the extraction of bioactive peptides important for the body.
The section also describes data relating to the part of the blood called "cruer", which is 90% haemoglobin. It is this part, after enzymatic hydrolysis, that can be a good source of valuable peptides, such as the α137-141 fragment.
It has also been described that the blood of vertebrates is a blood plasma rich in proteins and cruor, of which a large percentage is hemoglobin. Hemoglobin accounts for more than half of the proteins in many databases that refer to the various activities associated with active peptides.
Information is also given on the method and stages of enzymatic hydrolysis of hemoglobin in cattle and humans in order to obtain active peptides with valuable functions and applications.
In the "Results and Discussion" section, an In silico comparative analysis of bovine (Bos taurus) and human (Homo sapiens) hemoglobin is made.
The two alpha chains of BosTaurus and Homo sapiens (Figure 2A, SIM program) were found to have many similarities in their amino acid sequences: 125 out of 142 are strictly ordered; the obtained identity value is 88%. Considering similar amino acids, this positive value increases by up to 92%. This arrangement of the alpha chains of hemoglobin allows to observe that the peptides GAEALER (haematopoietic) and TSKYR (opioid, antimicrobial and antioxidant) are conserved. In contrast, the peptides LANVST (analgesic and bradykinin potentiator), ANVST (DPP-IV inhibitor) and KLLSHSL (antihypertensive) are present only in the hemoglobin of Bos Taurus.
The identity between the β-strands of BosTaurus and Homo sapiens has also been reported to be 85%, rising to 91% when similar amino acids are considered. The peptides VVYP (lipid-lowering) and LVVYPWTQRFF (opioid) are conserved, as shown by this arrangement of the hemoglobin β-chains. In addition, it should be noted that the human hemoglobin peptide VAGVANALAHKYH (coronary constrictor) is modified to the peptide VAGVANALAHRYH in the β-chain of bovine hemoglobin. The STPDA peptide in Bos taurus hemoglobin is modified into a STADA (bacterial growth stimulant) peptide in the human hemoglobin chain.
A Bioinformatics approach was also applied to predict peptides obtained from the pepsic hydrolysis of bovine hemoglobin Bos Taurus and human hemoglobin Homo sapiens.
Alpha and beta chain sequences of Bos taurus hemoglobin (GI: 116812902; GI: 27819608) and human hemoglobin (GI: 4504345; GI: 4504349) were reported and submitted to Peptide Cutter software (Expasy). When comparable amino acids are considered, Blast reports 92% identity between the alpha chains of Homo sapiens and Bos taurus. Peptide Cutter has identified 4 different cleavage sites and found that for the alpha chain of human hemoglobin, one of the missing cleavages is the one that produces the VAAA peptide (DPP-IV inhibitor) in bovine hemoglobin. This peptide is predicted not to be recovered during hydrolysis of human hemoglobin.
It is explained that of the 42 identical cleavage sites predicted by Peptide Cutter for peptic hydrolysis at pH higher than 2 on the hemoglobin beta chains of Bos Taurus and Homo sapiens, 10 sites are actually not identical.
The degree of hydrolysis was also determined by comparing the different DH under the applied hydrolysis conditions (23°C, pH 3.5) and at CBH; CHH of 1% (w/v).
It was found that over time the evolution of DH was identical, with the two curves having the same shape. The curves have the same shape as those previously reported in the literature. Accordingly, the DH was 2, 3, 4.5, 6, 8 and 10% when the hydrolysis time was 2.5, 5, 15, 30 min, 1, 2 and 3 h under these conditions.
The identification of the α137-141 peptide, which was performed with Waters software, is explained. This method has been shown to be effective in the identification and quantification of α137-141 in bovine cruor previously published in the literature
Standard α137-141 was noted to have a retention time of approximately 5.3-5.5 minutes. Peptides from human and bovine hydrolysates were then analyzed by comparing them to a standard spectrum. With the four parameters (MA, MT, PA and PT) and the retention time, the peak at a time just above 5 min was identified as α137-141. All measurements showed that PA was lower than PT and MA was lower than MT, highlighting the good purity of the peak and the similarity of the spectrum to α137-141.
A comparison of second derivative spectra between standard α137-141 peptide (a) and α137-141 peptide obtained from the hydrolysis of bovine hemoglobin (b) or human hemoglobin (c) by RP-UPLC was made. This analysis allows assessment of the presence or absence of aromatic amino acids in the peak.
In this case, the presence of a tyrosine corresponding to the sequence of the α137-141 peptide was assessed. The specific maxima at 288.7 nm indicate the presence of this amino acid. Their excellent similarity thus allowed peaks at a retention time of 5 min to be identified in the human and bovine hydrolysates as α137-141.
In this study, the reaction mechanism of enzymatic hydrolysis of human hemoglobin by pepsin was also investigated. UPLC chromatographic profiles obtained as a function of time are analyzed.
Obtained at identical hydrolysis times with bovine hemoglobin on the one hand and human hemoglobin on the other hand, both CBH and CHH equaled 1% (w/v) at 23°C, pH 3.5. At first glance, it can be noticed that for a given DH, the chromatograms were almost identical. This element ensures that the kinetics of the production of peptides from bovine hemoglobin and that of man were subject to the same reaction mechanisms.
Before the addition of pepsin, three peaks were detected at T0: α-chain, β-chain and heme. The hemoglobin subunits were rapidly hydrolyzed after the addition of the enzyme, resulting in intermediate peptides that were then converted to final peptides, which were generally more hydrophilic and smaller than the intermediate peptides. This is explained by the fact that hemoglobin (bovine or human) was originally in its natural form, called "globular". The latter was denatured by lowering the pH to 3.5 by adding 2M HCl. Thus it is found in an acidic environment in its "expanded" form. This favors the attack of the alpha and beta chains by the enzyme. This favors the "Zipper" enzymatic mechanism to the detriment of the "one-by-one" mechanism observed at higher pH [31]. At the end of the reaction, a set of peptides in the solution was analyzed.
The analysis found that the hydrolysis of human hemoglobin showed the same enzymatic mechanism as the hydrolysis of bovine hemoglobin.
The preparation of the active peptide α137-141 during the peptide hydrolysis of human and bovine hemoglobin is also described in this paper. An effect of increasing the initial concentration of bovine hemoglobin was applied to affect the enzyme and obtain an active peptide α137-141. Regardless of the initial concentration of human/bovine hemoglobin used, the zipper-type mechanism of hydrolysis remained identical.
It has been found that when a hemoglobin concentration higher than 10% (w/v) is used, a problem can arise because in this case the initial medium is no longer homogeneous if the hemoglobin dissolution is simply done in water, without adding salt.
When quantifying the production of the active peptide α137-141 from human and bovine hemoglobin, similarities were found in the way of obtaining the final amounts of this peptide, that is, the mechanism of hydrolysis is identical.
A peptidomic approach was also applied to characterize the peptide populations. Peaks® Studio XPro identified 217 and 189 unique peptides from bovine and human hemoglobin hydrolysates, respectively. Among the latter, only 57 are unique peptides strictly distributed among species.
In the section "Material and method" the method of comparison of peptide sequences by a bioinformatics approach is described. Protein Basic Local Alignment Search Tool (BLAST®) software was used.
The BLAST algorithm used is Smith Waterman, which allows fast alignment while maintaining good sensitivity.
The section also describes the method of predicting cleavage sites using a bioinformatic approach, as well as all materials: used reagents, solvents and standards. There is also a detailed description of the preparation of the basic solutions of bovine and human hemoglobin, the carrying out of the enzymatic hydrolysis of the two solutions and the determination of the degree of hydrolysis of the studied hemoglobin types. The hardware, software and protocol used are also indicated. The method of identifying the α137-141 peptide, the implementation of a peptidomic approach, the preparation of two-dimensional (2D) and three-dimensional (3D) heat maps and the performed RP-UPLC statistical analysis are also described.
In the Conclusion section, it is noted, Hydrolysis of human hemoglobin follows the same enzymatic mechanism as bovine hemoglobin, allowing rapid production of the antimicrobial peptide α137-141 with peak concentration reached in only 3 hours, with over 63% generated in 30 minutes. By identifying bioactive peptides performed by peptidomics, it was observed that most of the active peptides (especially antimicrobial peptides) identified in cattle are also present in humans without modification or belong to the same family due to the presence of common active sites.
It is also reported that this study also reveals the presence of new bioactive peptides in human hemoglobin, including antibacterial peptides, opioid peptides, ACE inhibitor, anticancer agent, and antioxidant. Some of these peptides were already known, but others had never been reported before in human hemoglobin.
1. Overall opinion
The manuscript may be of potential interest to researchers, biochemists, and medical professionals, but needs thorough revision before publication to ensure better structure and flow. The text should be revised and organized. A more detailed discussion of the results is needed. More authors need to be cited to support the results obtained.
The English format needs revision regarding word forms and typos in the text.
2. Comments
The summary is too short. The abstract does not contain enough information about the analysis methods used in determining peptide sequences, predicting peptide cleavage sites, enzymatic hydrolysis of both solutions, the degree of hydrolysis of the studied hemoglobin species, the way of identifying the α137-141 peptide, the realization of a peptidomic approach, the preparation of two-dimensional (2D) and three-dimensional (3D) heat maps, and RP-UPLC statistical analysis.
A more detailed discussion of the results related to the comparative analysis of the bovine and human hemoglobin peptide sequences, the hydrolysis steps and the method of identification of the α137-141 peptide is needed. The obtained results should be compared with more similar studies.
The conclusion is short and does not contain numerical data regarding the obtained values of the investigated indicators of the two types of hemoglobin.
There is a lack of ideas for future research related to the topic.
The "Results and Discussion" section should be divided into the "Results" and "Discussion" sections separately.
To enlarge the spectra contained in Figure 5.
It is necessary to indicate the degree of credibility of the obtained results when they are indicated and discussed in more places in the text.
In the references of lines 749, 751, 753, 754, 758, 761, 764, 767, 771, 773, 776, 778, 781, 783 and all other references to the end, the names of the specified journals should be italicized.
Line 769 of references is missing information about the specified document, such as pages, document link, etc.
The English format needs revision regarding word forms and typos in the text.
Author Response
Reviewer 3
Comments and suggestions for authors
1: The summary is too short. The abstract does not contain enough information about the analysis methods used in determining peptide sequences, predicting peptide cleavage sites, enzymatic hydrolysis of both solutions, the degree of hydrolysis of the studied hemoglobin species, the way of identifying the α137-141 peptide, the realization of a peptidomic approach, the preparation of two-dimensional (2D) and three-dimensional (3D) heat maps, and RP-UPLC statistical analysis.
Answer: Changes made
2: A more detailed discussion of the results related to the comparative analysis of the bovine and human hemoglobin peptide sequences, the hydrolysis steps, and the method of identification of the α137-141 peptide is needed. The obtained results should be compared with more similar studies.
Answer: Changes made
- The conclusion is short and does not contain numerical data regarding the obtained values of the investigated indicators of the two types of hemoglobin. There is a lack of ideas for future research related to the topic.
Answer: Change made
4.. The "Results and discussion" section should be divided into "Results" and "Discussion" sections separately.
Answer: Changes made
- To enlarge the spectra contained in Figure 5.
Answer: Change made
- It is necessary to indicate the degree of credibility of the obtained results when they are indicated and discussed in more places in the text. In the references of lines 749, 751, 753, 754, 758, 761, 764, 767, 771, 773, 776, 778, 781, 783 and all other references to the end, the names of the specified journals should be italicized.
Answer: Changes made
Line 769 of the references is missing information about the specified document, such as pages, document link, and so on.
Answer: Change made
Round 2
Reviewer 1 Report
The author should answer two more questions:
why they explore human hemoglobin as source of bioactive peptides?
Does human hemoglobin have some special advantages compared to bovine hemoglobin?
Author Response
To Editor Office
International Journal of Molecular Sciences
Special Issue : “Antioxidants and Anti-Inflammatory, Antiproliferative Activities of Natural Products”
Dear Editor,
Please find enclosed a manuscript entitled “Potential of human hemoglobin as a source of bioactive peptides: comparative study of enzymatic hydrolysis with bovine hemoglobin and production of active peptide α137-141” by Ahlam Outman, Barbara Deracinois, Christophe Flahaut, Mira Abou Diab, Bernard Gressier, Bruno Eto and Naïma Nedjar, we would like to submit for publication in the special issue. This study highlights the potential of human hemoglobin as a source of bioactive peptides useful for the food or pharmaceutical industry.
The manuscript has been fully revised according to the comments and suggestions of the three reviewers have been taken into account. All corrections are in purple color to allow reviewers to check the changes.
Thank you for considering this manuscript for publication.
Yours sincerely,
Prof. Bruno ETO, PhD.
Guest Editor Special Issue
Laboratoires TBC, Faculty of Pharmaceutical and Biological Sciences, Lille
etobr@laboratoires-tbc.com
Responses to reviewer 2
Why they explore human hemoglobin as source of bioactive peptides?
Response
There are several reasons for exploring human hemoglobin as a source of bioactive peptides. First of all, its higher availability: Indeed, hemoglobin comes directly from the blood. It represents more than 90% of the cruor (the main component responsible for the red color of mammalian blood) (Bah et al., 2013). It is important to note that human blood is abundant and available, unlike saliva, tears or sweat are also a valuable source of active peptides but poses the problem of availability. This provides easy access to this resource for research and production of bioactive peptides. In addition, the bioactive activities of human hemoglobin are well known and the potential for discovering new peptides is real: Human hemoglobin is a protein that can be a valuable source of many bioactive peptides with various functions. It has been identified as a source of peptides acting as opioids, analgesics, hematopoietic agents, coronary constrictors, antigonadotrophic, antibacterial and many others. Thus, it contributes to the maintenance of tissue homeostasis (Ivanov et al., 1997; Song., 2013, Liepke et al. 2003 Mak et al., 2004). By exploring human hemoglobin, it is also possible to discover new bioactive peptides as yet unknown. The endogenous peptides present in human hemoglobin are already optimized to perform their functions in humans, which increases the likelihood of finding effective peptides that are well tolerated by the human organism.
Does human hemoglobin have some special advantages compared to bovine hemoglobin?
Response
Advantages related to endogenous origin: these peptides offer the advantage of being already optimized by evolution to perform their specific functions in humans. This natural optimization increases the likelihood of discovering bioactive peptides that are both effective in their therapeutic actions and well tolerated by the human organism. By harnessing human hemoglobin as a source of bioactive peptides, it is possible to benefit from their natural compatibility with the human biological system, which can facilitate the development of safer and more effective drugs targeting various diseases and conditions. Reduced tolerance and immunogenicity: Endogenous peptide derivatives, such as those derived from human hemoglobin, are generally better tolerated and less immunogenic than peptides based on foreign antigens. This may reduce the risk of adverse effects and intolerance when using these peptides for therapeutic purposes (Guaní-Guerra et al., 2010; Felício et al.,2017) If we take the example of cancer, which is one of the major public health problems and conventional treatments can cause many undesirable side effects. Indeed, several scientific studies have shown that certain antimicrobial peptides contained in human hemoglobin have demonstrated anticancer properties. These include selective cytotoxic effects on cancer cells while preserving normal cells (Guaní-Guerra et al., 2010; Gaspar et al., 2013; Matteo Bosso et al., 2018). By exploiting the specific properties of these peptides, it is possible to design more effective treatments with fewer unwanted side effects.
Références :
Bah, C. S. F., Bekhit, A. E.-D. A., Carne, A. & McConnell, M. A. (2013). "Slaughterhouse blood: an emerging source of bioactive compounds". Comprehensive Reviews in Food Science and Food Safety, 12, 314-331.
Song CZ, Wang QW, Song CC. Erythrocyte-based analgesic peptides. Regul Pept. 2013 Jan 10;180:58-61. doi: 10.1016/j.regpep.2012.11.003. Epub 2012 Dec 7. PMID: 23220007.
Ivanov VT, Karelin AA, Philippova MM, Nazimov IV, Pletnev VZ. Hemoglobin as a source of endogenous bioactive peptides: the concept of tissue-specific peptide pool. Biopolymers. 1997;43(2):171-88. doi: 10.1002/(SICI)1097-0282(1997)43:2<171::AID-BIP10>3.0.CO;2-O. PMID: 9216253.
Liepke, Cornelia, Susann Baxmann, Cornelia Heine, Nicole Breithaupt, LudgerStändker, et Wolf-Georg Forssmann. 2003. « Human hemoglobin-derived peptides exhibit antimicrobial activity: a class of host defense peptides ». Journal of Chromatography B 791 (1‑2): 345‑56.
Mak, Paweł, Kinga Wójcik, ŁukaszWicherek, Piotr Suder, et Adam Dubin. 2004. « Antibacterial hemoglobin peptides in human menstrual blood ». Peptides 25 (11): 1839‑47.
Guaní-Guerra, E., Santos-Mendoza, T., Lugo-Reyes, S. O., & Terán, L. M. (2010). Antimicrobial peptides: general overview and clinical implications in human health and disease. Clinical immunology, 135(1), 1-11.
Felício MR, Silva ON, Gonçalves S, Santos NC, Franco OL. Peptides with Dual Antimicrobial and Anticancer Activities. Front Chem. 2017 Feb 21;5:5. doi: 10.3389/fchem.2017.00005. PMID: 28271058; PMCID: PMC5318463.
Gaspar D, Veiga AS, Castanho MA. From antimicrobial to anticancer peptides. A review. Front Microbiol. 2013 Oct 1;4:294. doi: 10.3389/fmicb.2013.00294. PMID: 24101917; PMCID: PMC3787199.
Bosso, M., Ständker, L., Kirchhoff, F., & Münch, J. (2018). Exploiting the human peptidome for novel antimicrobial and anticancer agents. Bioorganic & Medicinal Chemistry, 26(10), 2719-2726.
